# BlindSight: Harnessing Sparsity for Efficient Vision-Language Models

## Abstract

Large vision-language models (VLMs) enable joint processing of text and images. However, the inclusion of vision data significantly increases the prompt length, resulting in a longer time to first token (TTFT). This bottleneck can be mitigated by leveraging the inherent sparsity in the attention computation. Analyzing these attention patterns in VLMs when processing a series of images, we observe the absence of inter-image attention in a substantial portion of layers. Based on this, we propose **BlindSight**: an approach optimize multi-image VLM inference using an input-template-aware attention sparsity mask without runtime overhead. We utilize a dataset to derive a prompt-agnostic categorization for attention heads: Dense, Sink, Intra-Image, and Intra-Image+Sink. We further develop a Triton-based GPU kernel to leverage this sparsity. BlindSight achieves a 1.8-3.2$\times$ speedup in the attention computation (prompt length 36K-300K). We evaluate BlindSight using VLMs such Qwen2-VL, Qwen2.5-VL and Gemma 3; observing only a 0.78% absolute accuracy degradation on average for the evaluated multi-image understanding benchmarks.

## 1 Introduction

Vision-language models (VLMs) have shown impressive capabilities in processing visual and textual information together (Wang et al., 2024b; Bai et al., 2025; Meta AI, 2025; Gemma team, 2025; Chen et al., 2024b). This has opened up several avenues for application in fields such as autonomous driving (Hu et al., 2023; Tian et al., 2024), finance (Wang et al., 2023; Bhatia et al., 2024), medical research (Delbrouck et al., 2022; Hartsock & Rasool, 2024) and robotics (Black et al., 2024; Brohan et al., 2023). VLMs often incorporate pre-trained vision encoders, with a transformer-based multimodal processing module to process both language and vision tokens. The context length in VLM queries is often significantly higher than in text-only models due to the large number of tokens introduced by images and videos. Applications utilizing videos or high-resolution images would further increase the context length.

Attention dominates during inference for longer context lengths due to its quadratic computational complexity. The time to first token (TTFT) (i.e., prefill time) for such long prompts can be in the order of minutes and may even require distributed computation (Liu et al., 2024; Brandon et al., 2023). In Llama2-7B (Meta AI, 2023), the attention operation accounts for 70% of the total prefill time when processing an input of 64K tokens. The real-time application of VLMs is therefore limited by the increase in TTFT. An additional consequence of the increased context length is the substantial increase in the KV cache size during the decode phase, which further increases the system memory requirements. This impact is expected to only worsen with the test-time scaling paradigm (Snell et al., 2024; Yao et al., 2023; Muennighoff et al., 2025; Sadhukhan et al., 2025).

VLMs differ from LLMs in that their inputs typically consist of an interleaved sequence of images and text. The length of each image segment may vary depending on the image resolution and tokenization approach. Previous works have observed that attention matrices tend to be sparse (Zaheer et al., 2021; Beltagy et al., 2020). We also observe this phenomenon in VLMs as shown in Fig. 1a. This sparsity can be exploited to accelerate the prefill time by masking out computations corresponding to the sparse elements. Existing VLM-specific sparsity-based optimization techniques, such as MMInference (Li et al., 2025) and Look-M (Wan et al., 2024), rely on computing partial attention score-based metrics during inference to identify intra-modality sparsification opportunities,

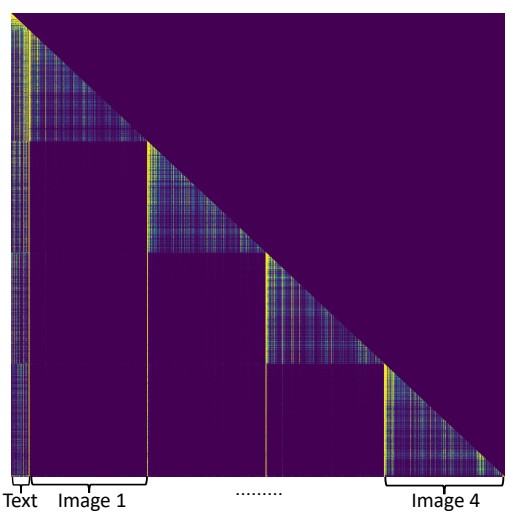 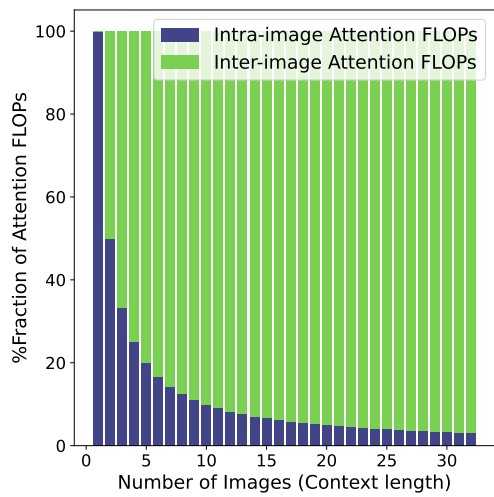

(a) Attention Score for Single Attention Head in VLM

(b) Intra-Image vs Inter-Image FLOPs

Figure 1: (a) Attention matrix for an attention head in Qwen2.5-VL (7B). The input prompt consists of text followed by 4 images. Notice that the image tokens vastly outnumber text tokens. (b) Impact of number of inputs images on intra/inter-image attention FLOPs.

at the cost of additional memory and latency. On the other hand, an offline sparsity characterization approach can remove such memory and latency overheads from the inference path.

We observe that inter-image attention dominates the overall attention computation as the number of images in the prompt increases (Fig. 1b). Based on this observation, we specifically focus on the *inter-image sparsity patterns in VLMs that can be derived without additional runtime computation*. We characterize the attention matrix for several multi-image VLMs such as Qwen2-VL (Wang et al., 2024b), Qwen2.5-VL (Bai et al., 2025) and Gemma 3 (Gemma team, 2025). We then categorize these sparsity patterns to derive distinct groups defined by the input prompt template (i.e., from the position of the text and images). We find that these patterns can be classified using a combination of intra-image and inter-image patterns: a non-sparse *Dense* pattern or a sparse *Sink*, *Intra-Image* or *Intra-Image+Sink* pattern (Section 3.2). We also observe an input-dependent variation in the optimal sparsity template selected per head. We determine that a carefully designed aggregation scheme can convert the prompt-dependent mask-template choice into a prompt-independent one (Section 4). **BlindSight** consists of two steps:

- **Prompt-level Characterization:** Metric-based approach to select the sparsity type for a given attention head with minimum accuracy impact.

- **Dataset-level Aggregation:** Rule-based approach to aggregate prompt-level attention head categorizations across a characterization dataset.

To our knowledge, BlindSight is the first technique that uncovers and leverages the image-sink based sparsity that emerges in VLMs processing multi-image prompts. We note the specific difference between multi-image and video processing in recent VLMs due to delimiter tokens (Section 7). Our contributions can be summarized as follows:

- Characterized attention patterns for several VLMs and identified four attention head categories: *Dense*, *Sink*, *Intra-Image* and *Intra-Image+Sink*.

- Developed **BlindSight**, an approach to select the optimal attention pattern per attention head in VLMs using an offline characterization flow.

- Implemented an efficient sparse attention kernel to leverage BlindSight's sparsity patterns.

- Investigated the underlying source of this sparsity, emphasizing the role of modality boundary tokens, which function as attention sinks per image.

## 2 RELATED WORK

**Static Attention Sparsity in LLMs:** Static post-training optimizations rely on offline characterization to identify fixed sparsity patterns to be used during inference. The sliding window attention layer introduced in LongFormers (Beltagy et al., 2020) computed the attention using only the most recent keys per query. Sparse Transformer (Child et al., 2019) employs a strided pattern along with windowing for efficient computation. Big Bird (Zaheer et al., 2021) enhances the sliding window approach by incorporating random blocks to capture block sparsity. Streaming LLM (Xiao et al., 2024b) highlighted the role of sink tokens, early tokens that garner high attention scores (Vig & Belinkov, 2019). This insight can be used to confine the computation to an A-shaped segment of the overall attention. While static sparsity patterns generally result in coherent conversations and efficient computation, they show poor performance on LLM benchmarks.

**Dynamic Attention Sparsity in LLMs:** Dynamic sparsity techniques rely on additional inference-time metrics to derive an input-dependent attention sparsity mask. DuoAttention (Xiao et al., 2024a) seeks to address gaps in static sparsity by classifying attention heads into streaming heads (A-shaped) and retrieval heads (full attention) through fine-tuning. SnapKV (Li et al., 2024) proposes compressing the KV cache by selecting out clustered KV positions for each head. MInference (Jiang et al., 2024b) proposes to categorize each head during inference into three types: A-shaped, block sparse and vertical-slash; though the vertical-slash pattern proves to be sufficient in practice. Furthermore, attention sparsity is also observed in the KV cache during the decode phase. Techniques such as H2O (Zhang et al., 2023), ScissorHands (Liu et al., 2023) and PyramidKV (Cai et al., 2025) prune past tokens using attention score-based metrics.

**Sparsity in VLMs:** MMInference (Li et al., 2025) dynamically derives intra-modality sparse patterns, similar to MInference, augmenting them with static cross-modality sparsity patterns derived for video inputs. The identification of the static sparsity patterns per head is conducted offline on a single sample. However, online computation is required to identify the intra-modality grid-like sparsity pattern. Look-M (Wan et al., 2024) optimizes the KV cache for VLMs using a KV cache merging scheme. VAR (Kang et al., 2025) identifies visual attention sinks in single-image scenarios.

## 3 ATTENTION SPARSITY IN VLMS

### 3.1 ATTENTION PRELIMINARIES

The scaled dot product attention utilized in the Multi-Head Attention layers (Vaswani et al., 2023) models the interactions of tokens within a sequence without relying on recurrence. We focus here specifically on the prefill stage. Given an input sequence of length $L$ and a model with hidden dimension $d_h$, linear projection layers first derive the query $Q$, key $K$ and value $V$ matrices; each with dimensions $L \times d_k$. The attention operation is then defined as:

$$\text{Attn}(Q, K, V) = \text{SoftMax}\left(\frac{QK^T}{\sqrt{d_k}} + Mask\right) V$$

During the prefill stage, the causal attention mask is used to prevent the interaction of a text token with future tokens. There is however a lack of standardization in the attention mechanism for the image component of a prompt, with either non-causal or causal masks used in different models. In this paper, we refer to a VLM's default attention mask as the Dense Mask. Per convention, $\text{SoftMax}\left(QK^T/\sqrt{d_k} + Mask\right)$ is referred to as the *attention matrix*.

### 3.2 SPARSITY IN VLMS

With a specific focus on inter-image interactions, we visually studied the attention matrix across layers for a variety of inputs. Although precise conclusions on sparsity masks can only be derived by studying the attention output, the attention matrix provides a visual proxy for analysis. Fig. 2 represents examples of the attention matrix for different types of sparse heads in Qwen2-VL (Wang et al., 2024b). We observe that these patterns remain consistent across other open-source VLMs such as Qwen3-VL, and Llama 4 (Appendix A). We observe that attention heads in VLMs belong to two categories, **Dense heads** and **Sparse heads**.

- **Dense Heads** have no discernible pattern of low-valued elements in the attention matrix, or consist of patterns that are not repeated across models and layers.

- **Sparse Heads** have many low-valued elements, and exhibit distinct boundaries at the text-image and image-image interface. They can be broadly categorized into three groups: **Sink Heads**, **Intra-Image Heads** and a hybrid **Intra-Image+Sink Heads**.

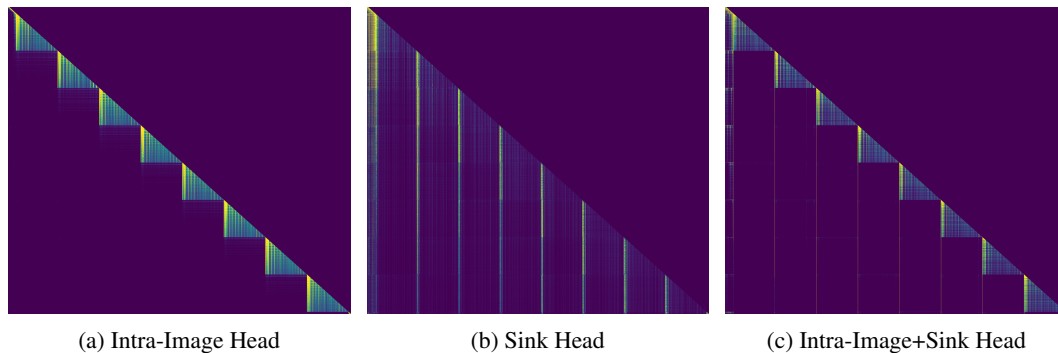

(a) Intra-Image Head     (b) Sink Head     (c) Intra-Image+Sink Head

Figure 2: Sparse attention categories in VLMs for prompts with multiple images

We further observe that variations exist in the sparsity patterns for sparse heads with different inputs. Further analysis of **Sparse heads** reveals that text-to-image or image-to-image transitions are frequently followed by an attention sink (Xiao et al., 2024b). **Sink heads** exhibit only this attention-sink behavior, with no intra-image attention. **Intra-Image heads** only attend within each image, with no sink-based cross-image attention. These heads focus mostly on intra-image processing. **Intra-Image+Sink heads** combine both the attention sink and intra-image pattern per image. Across all these scenarios, the text tokens continue to attend to all the previous tokens, leading to horizontal and vertical bands in the attention matrix. These sparsity categories offer a template for creating a mask based on the positions of the text and image tokens in the prompt. Note that the model's original masking approach (dense attention) can be used to handle non-sparse dense heads.

We exploit the inherent sparsity in VLMs by masking computations in attention heads that contribute negligibly to the final attention output. BlindSight optimizes inter-image attentions, and its impact is particularly relevant in multi-image scenarios (Figure 1b). The impact of optimizing intra-image attentions (Li et al., 2025) appears relatively modest for longer contexts dominated by inter-image attentions. We specifically focus on inputs containing multiple images rather than videos, as models typically lack per-frame delimiters for videos, crucial for VLM sparsity (Section 7).

## 4 BLINDSIGHT

BlindSight consists of two steps: prompt-level characterization and dataset-level aggregation. The first step involves characterizing attention heads for a single prompt. Here, we aim to identify the optimal sparsity pattern from the four categories. Given variations in the mapping for different prompts, we derive a single attention head to attention-type mapping across a characterization dataset by aggregating prompt-level categorizations.

### 4.1 PROMPT-LEVEL CHARACTERIZATION

Although visual characterization can provide valuable insights, practical deployments require an algorithmic approach to determine the optimal attention sparsity pattern. BlindSight addresses this need by minimizing the difference between the baseline dense attention output and the proposed sparse attention at every layer. Algorithm 1 describes the approach in detail. Every sparsity pattern is associated with a sparsity mask to compute the attention. Aligning the outputs at each layer ensures that the overall model remains aligned. A threshold $\alpha_{layer}$ on the normalized mean squared error (NMSE) serves as the selection criterion. The algorithm prioritizes patterns that achieve the lowest theoretical FLOPs. We revert to the original dense pattern when no sparse attention achieves an NMSE below $\alpha_{layer}$.

---

**Algorithm 1** BlindSight: Prompt-level Characterization

---

**Input**: layer; head; $Q, K, V \in \mathcal{R}^{L \times d_k}$; $\alpha_{layer}$
**Output**: head_type[layer][head]

$\text{Attn}_{\text{ref}} = \text{SoftMax} \left( \frac{QK^T}{\sqrt{d_k}} + \text{DenseMask} \right) V$
head_type[layer][head] = 'Dense Head'
**for** mask $\in$ ['Sink Head', 'Intra-Image Head', 'Intra-Image+Sink Head'] **do**
    $\text{Attn}_{\text{mask}} = \text{SoftMax} \left( \frac{QK^T}{\sqrt{d_k}} + \text{mask} \right) V$
    $\text{NMSE}_{\text{mask}} = ||\text{Attn}_{\text{mask}} - \text{Attn}_{\text{ref}}||_2^2 / ||\text{Attn}_{\text{ref}}||_2^2$
    **if** $\text{NMSE}_{\text{mask}} < \alpha_{layer}$ **then**
        head_type[layer][head] = mask
        **break**
    **end if**
**end for**
return head_type[layer][head]

---

## 4.2 DATASET-LEVEL AGGREGATION

We repeat the prompt-level characterization on a dataset to investigate the variation in masks selected for different prompts. For this characterization, we use the Multimodal Multi-image Understanding (MMIU) benchmark (Meng et al., 2024), which is designed to evaluate multi-image comprehension. We observe that most layers exhibit a preference for a single dominant attention pattern. In certain layers, however, we observe that the dense pattern is preferred for a non-negligible fraction, even though it is not the predominant choice. Detailed results on the sparsity pattern selected across layers are discussed in Appendix B.

We propose an empirical rule-based aggregation algorithm based on the distribution of the attention pattern across prompts. For every layer, we rely on the predominant attention pattern selected across the dataset, with the exception of instances where the Dense pattern exceeds a specified threshold fraction. This cautious strategy mitigates potential performance degradations associated with excessive sparsification. In scenarios where neither the Sink nor the Intra-Image pattern dominates, we revert to the Intra-Image+Sink category. As shown in Fig. 3, we observe that 60% the layers tend to be sparse across Qwen-type models.

---

**Algorithm 2** BlindSight: Dataset-level Aggregation

---

**Input**: layer; head; head_type_fraction; $\gamma_d$; $\gamma_s$; $\gamma_i$
**if** head_type_fraction[layer][head]['Dense'] $> \gamma_d$ **then**
    return 'Dense'
**end if**
**if** head_type_fraction[layer][head]['Sink'] $> \gamma_s$ **then**
    return 'Sink'
**end if**
**if** head_type_fraction[layer][head]['Intra-Image'] $> \gamma_i$ **then**
    return 'Intra-Image'
**else**
    return 'Intra-Image+Sink'
**end if**

---

## 5 BLINDSIGHT SPARSE ATTENTION KERNEL

We develop a Triton-based GPU compute kernel (Tillet et al., 2019) for BlindSight that exploits the sparsity patterns identified in VLMs. Following Flash Attention (Dao et al., 2022), the BlindSight kernel partitions the attention computation into tiles (contiguous chunks of queries, keys and values) and computes the attention within each tile. We implement four subroutines, each optimized for different scenarios within a tile. The primary distinction between subroutines lies in their mask

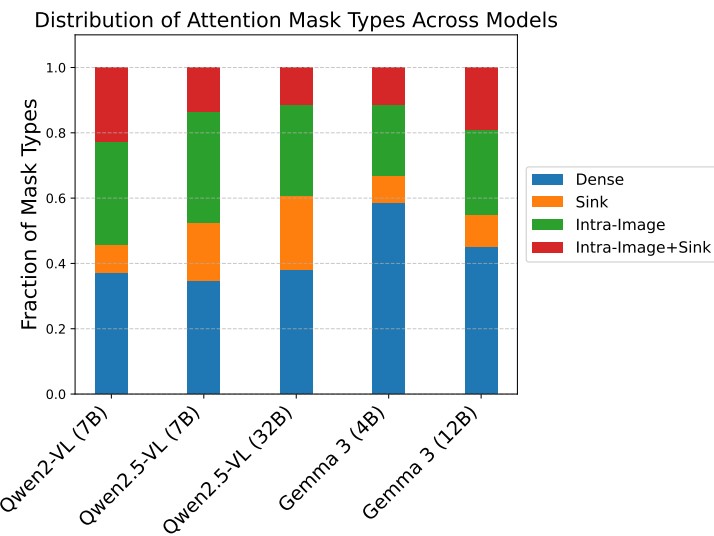

Figure 3: Distribution of sparsity categories across VLMs

construction logic. During runtime, the appropriate subroutine is selected based on the attention-head type and the presence of text or image tokens in the tile. The kernel receives the attention head type, the tile indices, and the text/image token boundaries (for the query, key dimensions) to construct the attention mask. Completely sparse tiles are skipped. Appendix L provides further details on the kernel, including subroutines and optimizations.

We demonstrate performance gains by integrating the BlindSight kernel into the Qwen 2.5-VL (7B) model (Setup per Section 6.1). Fig. 4a illustrates the latency improvements of the kernel compared to the Triton-based Flash Attention kernel over different sequence lengths, across only the attention layers. The kernel achieves increased speedup as the context length increases (i.e., as the number of images grows), due to increased sparsity. However, at very long sequence lengths, the $O(N^2)$ complexity of the dense heads begins to manifest itself through plateauing gains. Fig. 4b reveals the overall impact using the BlindSight kernel. The TTFT is improved $2.2\times$ at a sequence length of 300K. Bottlenecks in other modules masks the attention level gains. BlindSight exhibits a nearly linear trend in TTFT with sequence length, in contrast to the quadratic trend without sparsity.

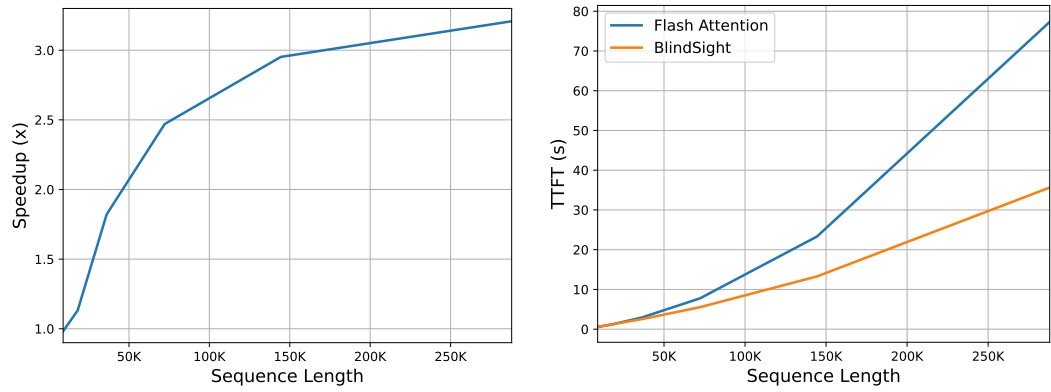

(a) BlindSight-based attention speedup over layers     (b) TTFT Comparison: Flash Attention vs BlindSight

Figure 4: BlindSight Kernel: Performance improvements in Qwen 2.5-VL (7B)

# 6 EVALUATION

We evaluate BlindSight on different VLMs using benchmarks for multi-image comprehension. This section first provides details on the experimental setup, models and benchmarks. Finally, we compare the accuracies of the original and sparsified models.

## 6.1 EXPERIMENTAL SETUP

**Models** We use open-source models available through Hugging Face (Wolf et al., 2020) to implement our proposed solution. The experiments focus on Qwen2-VL (7B) (Wang et al., 2024b), Qwen2.5-VL (7B, 32B) (Bai et al., 2025), and Gemma 3 (4B, 12B) (Gemma team, 2025). All of these models include a large transformer-based decoder module to jointly process vision embeddings (derived from a vision encoder) and text embeddings.

**Implementation** We utilize attention mechanisms specific to each model within the Hugging Face Transformers library (Wolf et al., 2020) to develop a custom attention mask variant. This custom sparse attention layer is integrated into the base model via monkey-patching. While the characterization stage uses a native PyTorch-based (Paszke et al., 2019) implementation, the BlindSight kernel is used for inference.

**Sparse Mask Generation** The sparse attention mask template requires knowledge of the positions of text and images within the prompt. These positions are determined by identifying the `<image_start>` and `<image_end>` tokens within the tokenized prompt. Each type of sparse mask (Sink, Intra-Image, Intra-Image+Sink) employs a different implementation based on these positions. The attention sink for every image is set to $10\%$ of the corresponding image length. Note that in this work, we only aim to optimize prefill time for VLMs and retain the entire KV cache.

The position of attention sinks in the sparse layers depends upon the specific implementation and training recipe used in the model. For typical VLMs such as the Qwen family, the attention sinks always occur at the start of the image. Gemma 3 tokenizes every image to the same length and employs a non-causal attention mechanism within an image. The attention sinks here occur at fixed locations within every image as shown in Fig. 8. We identified the locations of the attention sinks using the MMIU dataset (Meng et al., 2024), and selected the top-$10\%$ with the highest frequency.

**Infrastructure** The experiments were carried out on an AMD Instinct MI300X node with 8 GPUs. We utilized a PyTorch container with ROCm installed for development and experimentation. With 192 GB of HBM, we were able to load the entire model onto a single GPU for visual analysis and performance evaluation.

**Hyperparameter Selection** hyperparameters ($\alpha_{layer}$, $\gamma_d$, $\gamma_s$, $\gamma_i$) can be used to trade off sparsity and accuracy (Appendix D). For VLMs with causal intra-image attention displaying sparsity patterns similar to Fig. 2 (Qwen2-VL, Qwen2.5-VL, Qwen3-VL, Llama 4), we recommend using a fixed $\alpha_{layer} = 0.1$, $\gamma_d = 0.25$ and $\gamma_s = \gamma_i = 0.60$. We also recommend that users conduct experimental studies to trade off accuracy and performance to select an acceptable operating point (Appendix I). For atypical models such as Gemma 3 (non-causal intra-image attention), the sensitivity of the initial layers may affect performance. We use the empirically identified Linear $\alpha_{layer}$ scheme (Cai et al., 2025), i.e $\alpha_{layer} = 0.005 + (0.195 - 0.005) * \frac{layer_{idx}}{num_{layers}}$ for Gemma 3 (12B). Finally, when handling a smaller model with lower potential sparsity such as Gemma 3 (4B), we recommend reducing thresholds to preserve performance: $\alpha_{layer} = 0.001 + (0.099 - 0.001) * \frac{layer_{idx}}{num_{layers}}$.

## 6.2 PERFORMANCE EVALUATION

BlindSight is a prefill optimization strategy that specifically impacts scenarios involving image-to-image interactions. We therefore evaluate BlindSight using a variety of multi-image understanding benchmarks. These benchmarks typically present a sequence of images accompanied by a multiple-choice question pertaining to the image content. The MMIU (Meng et al., 2024), MuirBench (Wang et al., 2024a), MANTIS-eval (Jiang et al., 2024a), MIRB (Zhao et al., 2024), MMT (Ying et al., 2024) and MMIE (Xia et al., 2024) benchmarks are used to assess VLM comprehension. For the MMIE benchmark, we specifically evaluate the multi-step reasoning (MSR) task. Additional details on the benchmarks have been discussed in Appendix E.

**Qwen2-VL (7B)**

|  | MMIU | MANTIS | MuirBench | MIRB | MMT | MMIE |
|---|---|---|---|---|---|---|
| Original | 34.12 | 61.61 | 49.83 | 70.77 | 63.17 | 70.20 |
| All Sink Heads | 33.61 | 60.19 | 42.57 | 66.89 | 61.58 | 67.12 |
| All Intra-Image Heads | 34.90 | 57.35 | 43.95 | 67.83 | 62.67 | 69.91 |
| All Intra-Image+Sink Heads | 34.12 | 62.56 | 47.45 | 69.22 | 62.69 | 70.05 |
| BlindSight ($\alpha_{layer} = 0.1$) | 34.90 | 63.03 | 48.86 | 69.61 | 63.13 | 70.00 |

**Qwen2.5-VL (7B)**

|  | MMIU | MANTIS | MuirBench | MIRB | MMT | MMIE |
|---|---|---|---|---|---|---|
| Original | 35.51 | 68.25 | 55.31 | 71.12 | 61.29 | 71.25 |
| All Sink Heads | 30.54 | 62.56 | 47.29 | 69.21 | 60.41 | 67.93 |
| All Intra-Image Heads | 31.83 | 54.98 | 47.64 | 69.98 | 61.22 | 70.13 |
| All Intra-Image+Sink Heads | 34.23 | 62.09 | 51.33 | 70.01 | 61.27 | 70.21 |
| BlindSight ($\alpha_{layer} = 0.1$) | 35.06 | 66.35 | 53.75 | 70.30 | 61.35 | 70.83 |

**Qwen2.5-VL (32B)**

|  | MMIU | MANTIS | MuirBench | MIRB | MMT | MMIE |
|---|---|---|---|---|---|---|
| Original | 44.67 | 72.51 | 61.87 | 73.37 | 65.85 | 78.28 |
| All Sink Heads | 35.60 | 64.93 | 42.80 | 68.89 | 63.04 | 74.20 |
| All Intra-Image Heads | 38.19 | 58.77 | 49.14 | 71.9 | 65.21 | 76.92 |
| All Intra-Image+Sink Heads | 37.58 | 67.77 | 56.19 | 72.11 | 65.32 | 77.21 |
| BlindSight ($\alpha_{layer} = 0.1$) | 44.10 | 70.62 | 61.64 | 72.39 | 65.42 | 77.80 |

**Gemma 3 (4B)**

|  | MMIU | MANTIS | MuirBench | MIRB | MMT | MMIE |
|---|---|---|---|---|---|---|
| Original | 33.78 | 63.51 | 35.39 | 53.83 | 54.63 | 60.96 |
| All Sink Heads | 29.44 | 46.08 | 24.23 | 35.22 | 43.46 | 46.05 |
| All Intra-Image Heads | 34.44 | 49.77 | 29.77 | 40.17 | 54.56 | 58.52 |
| All Intra-Image+Sink Heads | 33.17 | 50.69 | 27.83 | 42.67 | 52.33 | 57.48 |
| BlindSight (Linear $\alpha_{layer}$) | 35.01 | 60.19 | 33.93 | 51.34 | 54.76 | 60.93 |

**Gemma 3 (12B)**

|  | MMIU | MANTIS | MuirBench | MIRB | MMT | MMIE |
|---|---|---|---|---|---|---|
| Original | 35.81 | 69.67 | 50.64 | 54.97 | 58.73 | 62.93 |
| All Sink Heads | 27.61 | 55.76 | 29.65 | 39.66 | 43.46 | 47.17 |
| All Intra-Image Heads | 33.67 | 57.58 | 41.19 | 43.9 | 57.95 | 59.80 |
| All Intra-Image+Sink Heads | 32.61 | 60.37 | 37.88 | 44.59 | 56.09 | 57.00 |
| BlindSight (Linear $\alpha_{layer}$) | 33.95 | 68.25 | 46.78 | 53.76 | 59.83 | 62.62 |

Table 1: Accuracy (%) on multi-image comprehension benchmarks

We compare the performance of BlindSight with sparse models in which every attention head is replaced with a sparse head category. The results shown in Table 1 highlight the minimal performance degradation of sparsifying a model using BlindSight. Typically, BlindSight's accuracy degradations are marginal, suggesting that a large portion of masked-out computations have minimal impact on the final prediction.

We note that a model comprising a mixture of sparse and dense attention heads, derived using Blind-Sight, consistently outperforms an all-sparse model. This suggests that full inter-image attention in dense heads and partial inter-image attention through sinks is sufficient to maintain performance on complex multi-image comprehension tasks. Among the sparse model evaluations, models with only attention sinks perform consistently worse, Finally, we also observe that BlindSight offers robustness to characterization datasets beyond MMIU (Appendix F), while also maintaining consistent performance across image tokenization lengths (Appendix G).

## 7 DISCUSSION

Attention sinks emerge in LLMs during pre-training, where the SoftMax operator is repeatedly applied on the early tokens (Xiao et al., 2024b). Additionally, massive activations occur independent of the input data within specific attention layers. These activations develop into sinks that function as implicit bias terms within the attention layer (Sun et al., 2024). Furthermore, delimiter tokens (punctuation,\n) with low semantic value often correspond to high attention scores Clark et al. (2019). Recent evidence (Gu et al., 2025; Barbero et al., 2025) highlights the crucial role of the attention sink in long-context learning.

We now empirically study the role of attention sinks associated with every image. VLM prompts typically consist of delimiter tokens at the start (`<image_start>`) and end (`<image_end>`) of every image. These image boundary tokens are essential for enabling sparsity. To support this claim, we study attention patterns removing these boundary tokens in a text-image interleaved prompt in Qwen2-VL (7B). We observe that the intra-image attention ceases to exist (Fig. 5). Partial information pooling across images still persists via weak sinks, resulting in incoherent responses.

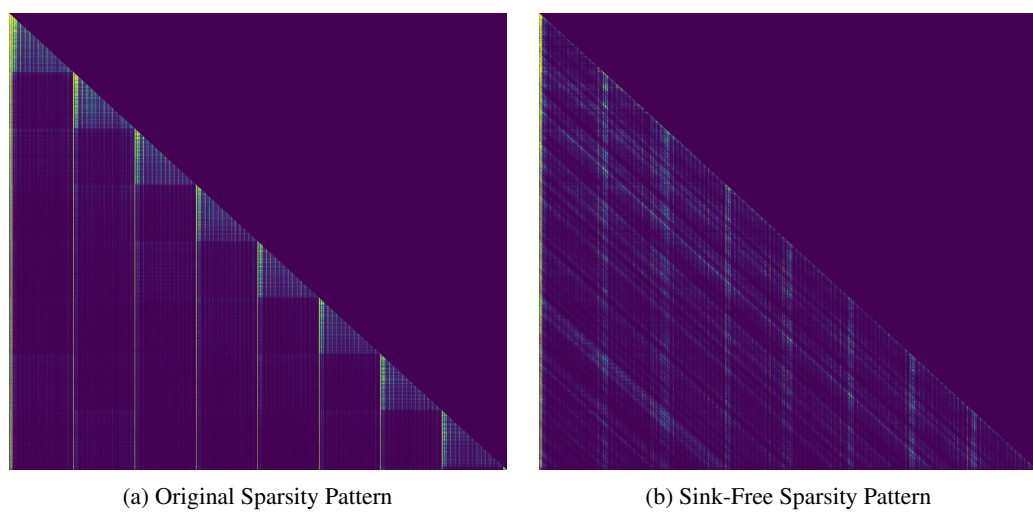

(a) Original Sparsity Pattern        (b) Sink-Free Sparsity Pattern

Figure 5: Impact of removing image boundary tokens: Removing `<image_start>` and `<image_end>` impairs attention sinks and disrupts the intra-image masking pattern.

In typical pre-processing schemes, videos are decomposed into images, with the entire video segment in the prompt enclosed by a single pair of delimiter tokens. In multi-image settings however, each image is enclosed by its own pair of delimiter tokens. Despite multi-image and video inputs being similar, models typically produce different sparsity patterns due to the lack of per-frame delimiter tokens. In line with our observation, Qwen3-VL recently introduced per frame delimiter tokens along with temporal markers, which induces attention sparsity for even video inputs (Appendix K).

While BlindSight is a post-training VLM optimization technique, we advocate for incorporating such sparsity during training into the model architecture itself. Llama 4 (Meta AI, 2025) includes chunked attention (document masking) layers, which also enable linear scaling in complexity at long sequence lengths. Native Sparse Attention (Yuan et al., 2025) and DeepSeek Sparse Attention (DeepSeek-AI, 2025) directly incorporate dynamic sparsity into the attention mechanism. Performant models can be composed through a mixture of low-complexity attention layers and dense attention layers, as evidenced by Table 1. Post-training sparsification (Xiao et al., 2024a) and hybrid state-space based models (Glorioso et al., 2024; Ren et al., 2025; Yang et al., 2025) highlight the potential to achieve improved efficiency and performance by strategically combining sparse, linear and dense attention mechanisms.

## 8 CONCLUSION

We investigated inter-image interactions within the attention layers of VLMs. From this analysis, we developed BlindSight: an input-template-based VLM sparsification technique that achieves performance levels comparable to the original dense attention across various multi-image understanding benchmarks. We developed a Triton-based kernel that showcased performance gains. We finally presented an empirical analysis on the origin of sparsity in VLMs with multi-image inputs. We anticipate that these findings will motivate future research to natively integrate hybrid sparse and dense attention mechanisms into VLM architectures.

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

# A    ATTENTION SPARSITY PATTERNS FOR ADDITIONAL MODELS

While we evaluated the performance of BlindSight on Qwen2-VL, Qwen2.5-VL and Gemma in this paper; we expect that BlindSight can be applied to other VLMs as well. As shown in Fig. 6, we observe the presence of sparse attention layers in VLMs such as Qwen3-VL and Llama 4.

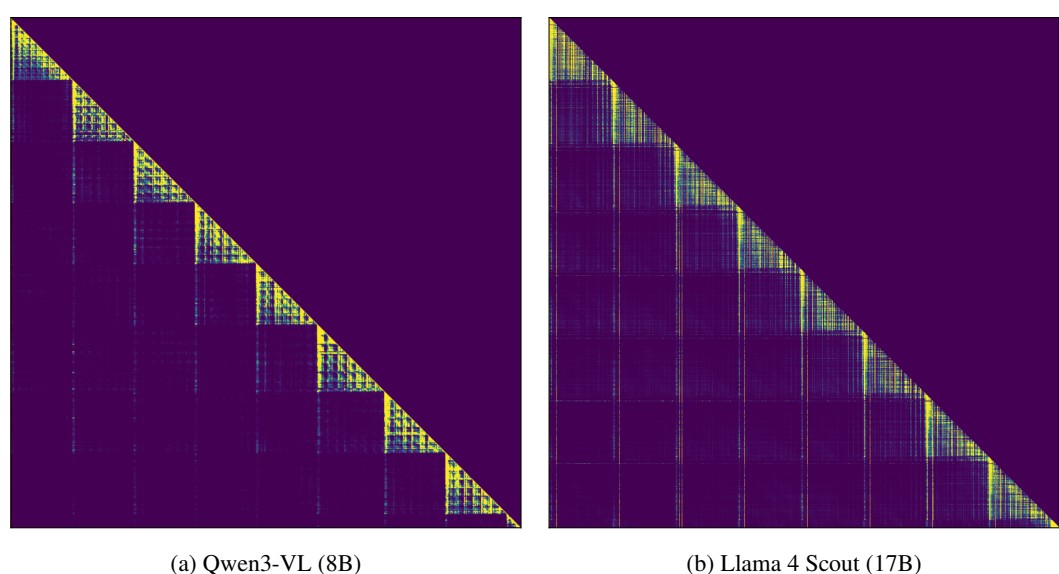

(a) Qwen3-VL (8B)                        (b) Llama 4 Scout (17B)

Figure 6: Sparse attention matrices observed in other VLMs

# B    SPARSITY ANALYSIS

We analyze the distribution of the head-types selected across a dataset using the prompt-level characterization scheme discussed in Algorithm 1. As observed in Fig. 7, a single mask category tends to dominate within a layer across multiple prompts. However, there tends to be a non-negligible fraction selecting other sparsity masks.

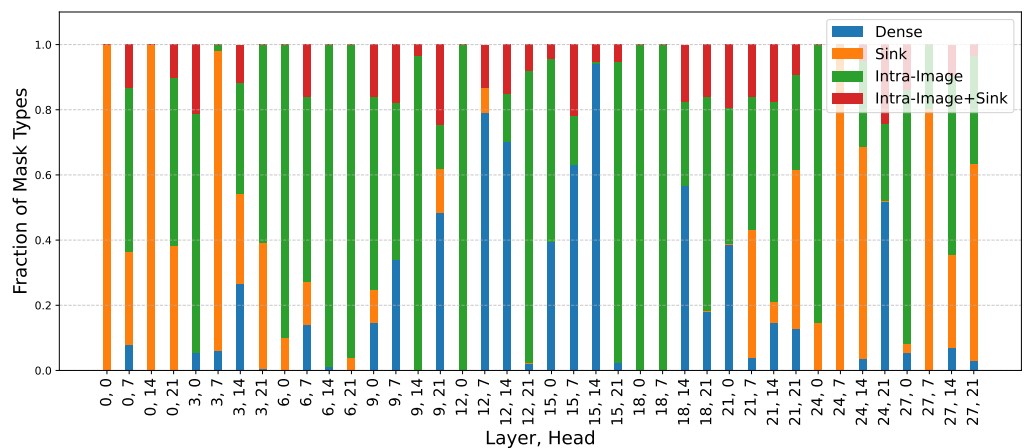

Figure 7: Distribution of sparse mask categories shown for select attention heads across the MMIU dataset using the prompt-level characterization scheme for Qwen 2.5-VL (7B).

## C  ATTENTION SINK POSITIONS IN GEMMA 3

Figure 8 represents the attention matrices for different attention heads in the Gemma 3 (4B) model. As shown in the figure, inter-image attention occurs mainly through attention sinks. However, these attention sinks are located at fixed offsets within the image as opposed to the image boundary in the case of Qwen models.

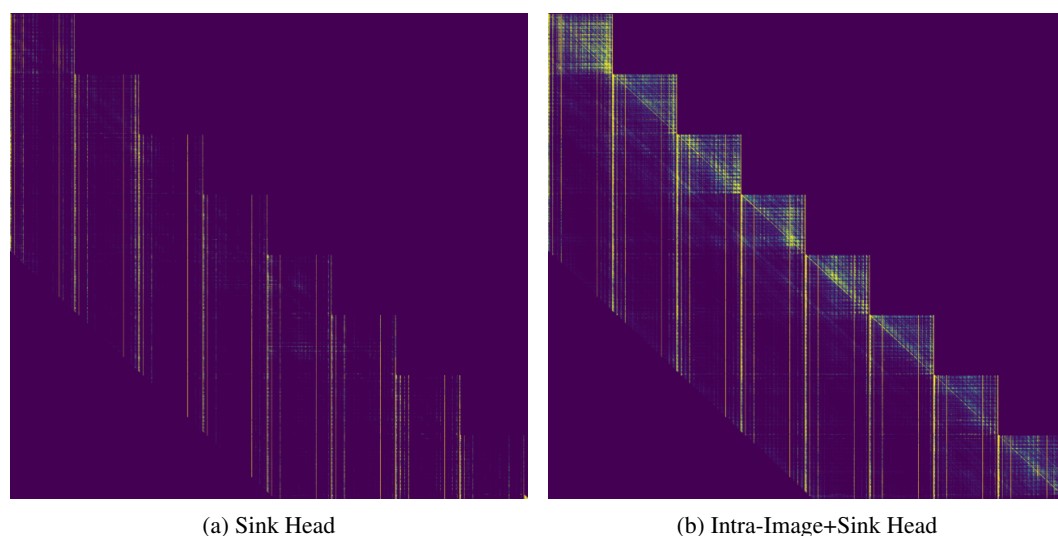

(a) Sink Head                                   (b) Intra-Image+Sink Head

Figure 8: Attention sink positions in Gemma 3

## D  ABLATION STUDIES FOR HYPERPARAMETERS

Configuring $\alpha_{layer}, \gamma_d, \gamma_s$ and $\gamma_i$ is essential for the optimal configuration of BlindSight. We ensure that the hyperparameters (apart from the one being varied) remain fixed to their default values of $\alpha_{layer} = 0.1, \gamma_d = 0.25$ and $\gamma_s = \gamma_i = 0.6$.

### D.1  IMPACT OF $\alpha_{layer}$ FOR PROMPT-LEVEL CHARACTERIZATION

$\alpha_{layer}$ sets the NMSE threshold above which the default dense pattern is selected for that particular prompt. A lower value of $\alpha_{layer}$ increases the occurrence of dense heads.

#### D.1.1  IMPACT OF FIXED $\alpha_{layer}$ FOR QWEN

Across the Qwen models, we fix $\alpha_{layer}$ across layers. A higher $\alpha_{layer}$ leads to a higher model sparsification (Fig. 9). Consequently, performance declines as the fraction of dense masks in the sparsified model is reduced.

| $\alpha_{layer}$ | 0.05 | 0.1 | 0.2 | 0.4 |
|---|---|---|---|---|
| MuirBench Accuracy (%) | 61.71 | 61.64 | 56.62 | 51.49 |

Table 2: Impact of selecting $\alpha_{layer}$ in Qwen2.5-VL (32B)

#### D.1.2  IMPACT OF LINEAR $\alpha_{layer}$ FOR GEMMA 3

Fig. 10a represents the attention head type distribution in the Gemma 3 model using a fixed $\alpha_{layer}$. In this setting, we observe that the initial layers have a higher number of sparse heads. Prior works (Cai et al., 2025) have shown that the initial layers are more sensitive to sparsification. Based on this, we employ a linearly increasing threshold $\alpha_{layer}$ to limit sparsification in the initial layers. Figure 10b represents the attention head distribution for linear $\alpha_{layer}$. As shown in the figure,

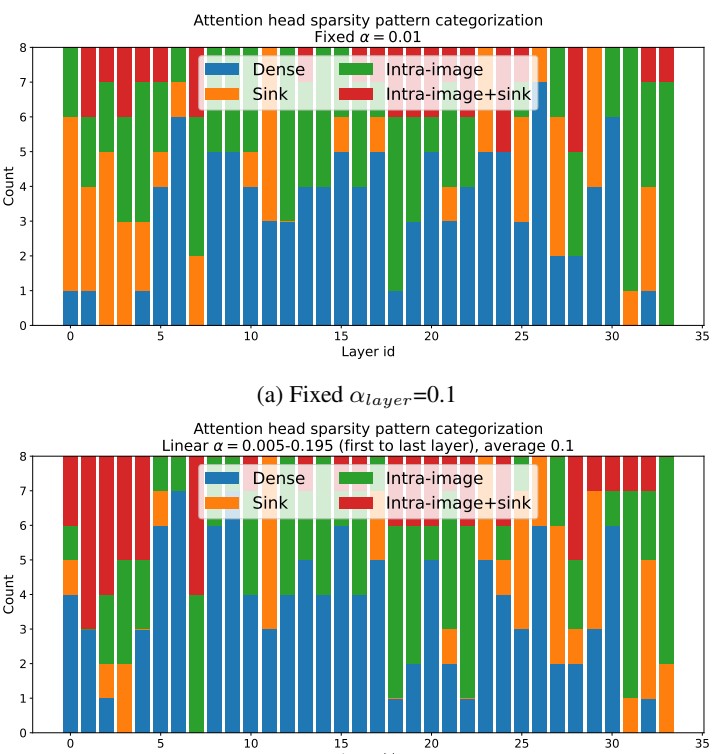

Figure 9: Attention head distribution in Qwen2.5-VL (32B) with different fixed $\alpha_{layer}$

(a) Fixed $\alpha_{layer}$=0.1

(b) Linear $\alpha_{layer}$ (0.005 to 0.195)

Figure 10: Attention head distribution for Gemma 3 (4B) with different $\alpha_{layer}$ strategies

this schedule promotes the selection of dense heads in the initial layers. We find that this strategy improves the end-to-end accuracy of the Gemma 3 (4B) model by 1-2% for several benchmarks. Table 3 compares accuracy for fixed and linear $\alpha$. Note that for the Qwen models, the accuracy does not change between the linear and fixed schemes.

### D.2    IMPACT OF $\gamma_d$, $\gamma_s$ AND $\gamma_i$ FOR DATASET-LEVEL AGGREGATION

$\gamma_d$, $\gamma_s$ and $\gamma_i$ are used in the Dataset-Level Aggregation step to merge individual sparsity mappings for different prompts. $\gamma_d$ is a threshold on the fraction of dense heads across prompts, used to qualify a given head as a dense head. A higher value of $\gamma_d$ leads to a configuration with a lower proportion of dense heads (Fig. 11a). $\alpha_{layer}$ and $\gamma_d$ control the presence of dense heads in the model.

| Gemma 3 (4B) | | | | |
| --- | --- | --- | --- | --- |
| | MMIU | MANTIS | MUIRBench | MIRB | MMT |
| Linear $\alpha$ | 635.01 | 60.19 | 33.93 | 51.34 | 54.76 |
| Fixed $\alpha$ | 34.78 | 58.99 | 33.58 | 46.5 | 54.37 |

| Gemma 3 (12B) | | | | |
| --- | --- | --- | --- | --- |
| | MMIU | MANTIS | MUIRBench | MIRB | MMT |
| Linear $\alpha$ | 33.95 | 68.25 | 46.78 | 53.76 | 59.83 |
| Fixed $\alpha$ | 33.83 | 68.2 | 45.81 | 49.74 | 58.55 |

Table 3: Accuracy (%) for fixed and linear $\alpha$ in Gemma 3 models

| $\gamma_d$ | 0.1 | 0.25 | 0.4 |
| --- | --- | --- | --- |
| MuirBench Accuracy (%) | 61.64 | 61.64 | 58.62 |

Table 4: Impact of varying $\gamma_d$ in Qwen2.5-VL (32B)

$\gamma_s$ and $\gamma_i$ only serve to control the distribution of masks selected among the different sparse masks. Intra-Image+Sink masks are a superset of intra-image and sink masks. Replacing Sink and Intra-Image heads with Intra-Image+Sink heads (Fig. 11b) is not expected to affect the accuracy.

| $\gamma_s, \gamma_i$ | 0.4 | 0.6 | 0.8 |
| --- | --- | --- | --- |
| MuirBench Accuracy (%) | 61.64 | 61.64 | 61.79 |

Table 5: Impact of varying $\gamma_s$, $\gamma_i$ in Qwen2.5-VL (32B)

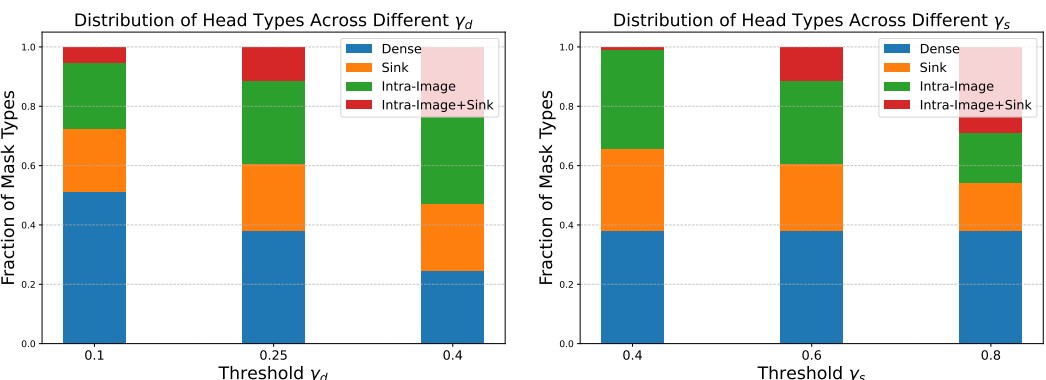

(a) $\gamma_d$: Dense Head fraction inversely correlated  (b) $\gamma_s$: Sink Head fraction inversely correlated

Figure 11: Head distribution with varying $\gamma_d$ and $\gamma_s$ in Qwen2.5-VL (32B)

# E    DETAILS ON MULTI-IMAGE COMPREHENSION BENCHMARKS

**MMIU** (Meng et al., 2024): Evaluates VLMs on multi-image prompts across multiple tasks encompassing 7 types of relationships such as semantic content, temporal relations and spatial understanding. This includes high-level tasks such as captioning, image ordering, similarity detection and text-to-image retrieval.

**MuirBench** (Wang et al., 2024a): Provides robust assessment through 12 diverse multi-image tasks spanning 10 categories of relations (2,600 multiple-choice questions with 11,264 images) such as

scene understanding, visual retrieval, geographic understanding, diagram understanding and ordering. This benchmark also includes unanswerable variants.

**MANTIS** (Jiang et al., 2024a): Evaluates multi-image model capabilities through a hand-curated dataset consisting of challenging image understanding questions comparing two images.

**MIRB** (Zhao et al., 2024): Benchmarks multi-image understanding across four evaluation dimensions: perception tasks (image jigsaw, counting, attribute matching), visual world knowledge (sightseeing locations, food comparisons), reasoning (code understanding, visual analogy, 3D scene understanding), and multi-hop reasoning (synthetic logic chains).

**MMT-Bench** (Ying et al., 2024): Assesses VLMs on 32 core competencies and 162 sub-tasks involving multi-image comprehension. This includes scenarios involving perception, summarization, ordering, retrieval, reasoning, visual coding and OCR detection.

**MMIE** (Xia et al., 2024): Evaluates VLMs on interleaved multimodal comprehension tasks across 20K curated queries spanning 12 fields and 102 sub-fields including mathematics, coding, physics, literature, health, and arts.

## F  ROBUSTNESS TO CHARACTERIZATION DATASET

We highlight the robustness of BlindSight to the characterization dataset in this section. Note that our previous experimental evaluations had utilized the MMIU (Meng et al., 2024) dataset. In Table 6, we observe that characterization on the MuirBench (Wang et al., 2024a) benchmark results in performance similar to the MMIU-based characterization. In general, we recommend the use of a multi-image dataset with diversity in prompt structure.

| Benchmark | MMIU | MANTIS | MuirBench |
|---|---|---|---|
| Original | 44.67 | 72.51 | 61.87 |
| BlindSight (MMIU Char.) | 44.10 | 70.62 | 61.64 |
| BlindSight (MuirBench Char.) | 43.90 | 70.33 | 62.06 |

Table 6: Accuracy (%) on MuirBench for Qwen2.5-VL (32B) w/ diff. char. datasets.

## G  ROBUSTNESS TO IMAGE TOKENIZATION LENGTH

Qwen-style models utilize a flexible image tokenization scheme. In our Qwen evaluation discussed in Table 1, we configure a range between 1280 and 5120 tokens/image. We now study BlindSight's performance with a fixed number of tokens per image. We observe that BlindSight's performance does not show significantly degrade on the MuirBench (Wang et al., 2024a) evaluation compared to the baseline (Table 7) across token lengths per image.

| | 320 | 640 | 1280 | 2560 | 5120 |
|---|---|---|---|---|---|
| Original | 62.14 | 62.83 | 62.29 | 61.13 | 61.10 |
| BlindSight | 61.75 | 61.79 | 61.38 | 60.67 | 60.53 |

Table 7: Accuracy (%) on MuirBench for Qwen2.5-VL (32B) w/ diff. image tokenization length

## H  FLOPS REDUCTION FOR EVALUATED BENCHMARKS

BlindSight results in savings in the attention FLOPs due to increased sparsity. We summarize the FLOPs reduction in the attention computation for all the evaluated benchmarks in Table 8. We eval-

uate these metrics on the MuirBench (Wang et al., 2024a) dataset. The table highlights the statistics ($Q_1$: 25th percentile, $Q_2$: Median, $Q_3$: 75th percentile) of the percentage of FLOPs saved across all attention layers in the model, computed across the various prompt structures in the benchmark. Note that while Qwen models allow for flexible tokenization per image (1280-5120 in our setup), Gemma models fix this to 128 vision tokens/image. With BlindSight masking primarily the inter-image component of the attention, a reduced fraction of FLOPs are saved on the Gemma models.

| **Qwen2-VL (7B)** | | | | | | |
|---|---|---|---|---|---|---|
| | MMIU | MANTIS | MuirBench | MIRB | MMT | MMIE |
| $Q_1$ | 26.69 | 28.52 | 40.72 | 17.51 | 35.94 | 31.01 |
| $Q_2$ (Median) | 39.18 | 29.16 | 45.25 | 41.84 | 44.00 | 31.46 |
| $Q_3$ | 47.07 | 37.98 | 47.98 | 46.28 | 46.86 | 32.96 |

| **Qwen2.5-VL (7B)** | | | | | | |
|---|---|---|---|---|---|---|
| | MMIU | MANTIS | MuirBench | MIRB | MMT | MMIE |
| $Q_1$ | 30.79 | 32.97 | 44.72 | 15.47 | 39.38 | 35.76 |
| $Q_2$ (Median) | 43.03 | 33.91 | 48.80 | 45.95 | 47.45 | 36.28 |
| $Q_3$ | 50.29 | 41.71 | 51.26 | 49.91 | 50.06 | 38.01 |

| **Qwen2.5-VL (32B)** | | | | | | |
|---|---|---|---|---|---|---|
| | MMIU | MANTIS | MuirBench | MIRB | MMT | MMIE |
| $Q_1$ | 31.12 | 33.47 | 43.84 | 19.75 | 39.52 | 36.15 |
| $Q_2$ (Median) | 42.19 | 34.31 | 47.29 | 45.04 | 45.97 | 36.97 |
| $Q_3$ | 48.42 | 40.96 | 49.35 | 48.36 | 48.20 | 38.41 |

| **Gemma 3 (4B)** | | | | | | |
|---|---|---|---|---|---|---|
| | MMIU | MANTIS | MuirBench | MIRB | MMT | MMIE |
| $Q_1$ | 9.06 | 11.50 | 28.76 | 32.38 | 19.55 | 25.71 |
| $Q_2$ (Median) | 23.51 | 12.19 | 31.81 | 35.09 | 27.09 | 29.08 |
| $Q_3$ | 30.22 | 17.82 | 33.20 | 36.04 | 28.71 | 30.64 |

| **Gemma 3 (12B)** | | | | | | |
|---|---|---|---|---|---|---|
| | MMIU | MANTIS | MuirBench | MIRB | MMT | MMIE |
| $Q_1$ | 11.91 | 14.98 | 37.83 | 42.59 | 25.70 | 25.77 |
| $Q_2$ (Median) | 30.93 | 15.88 | 41.84 | 46.15 | 35.64 | 29.08 |
| $Q_3$ | 37.83 | 23.29 | 43.66 | 47.40 | 37.77 | 30.64 |

Table 8: Attention FLOPs reduction (%) statistics across evaluated benchmarks

## I   ACCURACY-FLOPS TRADE-OFFS

BlindSight's hyperparameters control the distribution of the layer categories in the sparsified model. Increased sparsity is expected to result in lower TTFT, but at the cost of accuracy. This behavior can be observed studying the FLOPS savings across different $\alpha_{layer}$ (Appendix D.1.1). In Fig. 12, we compare the attention FLOPs savings on MuirBench (Wang et al., 2024a) for different sparsity levels. Note that variations in the prompt structure (number of images, image dimensions) lead to a distribution in FLOPs savings per $\alpha_{layer}$. Below a certain sparsification level ($\alpha_{layer} <= 0.1$), the performance of the sparse model saturates. We suggest that users conduct experimental studies on their end use-case to identify this knee point.

## J   COMPARISON WITH TOKEN PRUNING AND COMPRESSION TECHNIQUES

Several works have focused on different approaches for optimizing the inference and training runtime in VLMs; including token pruning/merging, model compression (quantization) and sparse at-

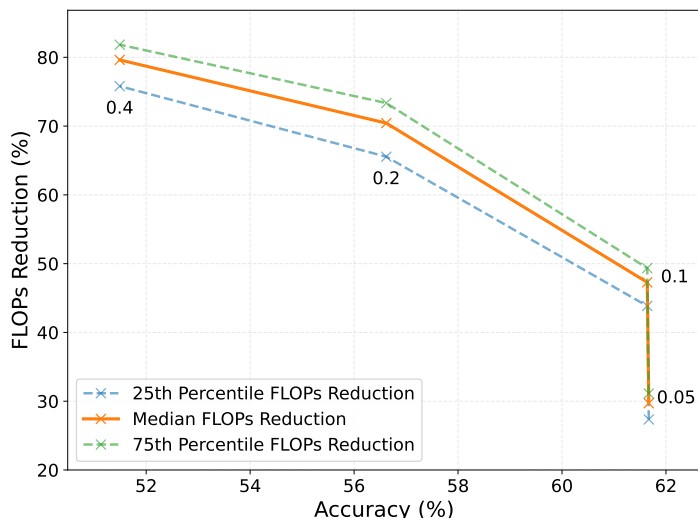

Figure 12: Distribution of attention FLOPS saved on the Qwen 2.5-VL (32B) on the MuirBench evaluation for varying $\alpha_{layer}$ (annotated).

tention. In this work, we specifically focus on sparsifying the attention matrix to reduce computation. BlindSight is an offline approach that does not result in any runtime overhead from online computation.

Another complementary approach to optimization is through token compression or pruning. Prior works such as Chen et al. (2024a); Alvar et al. (2025); Shang et al. (2024); Wen et al. (2025); Gong et al. (2024) have focused on using attention scores or token similarity to reduce the number of visual tokens by pruning or merging redundant tokens. Furthermore, existing token pruning and merging approaches have focused on single image or video benchmarks. Note that in this work, we specifically focus on multi-image inputs. We have shown previously that the inter-image attention component dominates the overall attention computation as the number of images in the prompt increase (Fig. 1b). The lack of evaluation and algorithm design based on multi-image benchmarks in existing token pruning/merging approaches makes it challenging to perform a meaningful quantitative comparison with BlindSight. We now highlight BlindSight's advantage against pruning with theoretical estimates and showcase it's compatibility with experimental evidence.

Figure 13 compares the theoretical attention FLOPs of token pruning and BlindSight. Note that the y-axis here is in a logarithmic scale here. The attention computation requirement with pruning still remains quadratic. However, BlindSight leverages inter-image and intra-image sparsity in VLMs; resulting in a sub-quadratic complexity.

We believe that token compression and BlindSight's sparse attention are orthogonal to each other. With this aim, we studied the behavior of DivPrune Alvar et al. (2025), a token pruning approach, on the Qwen2-VL (7B) model. We analyzed the attention matrices with BlindSight and BlindSight + DivPrune (90% pruning) in Fig. 14. We note that the original attention sparsity patterns are retained with token compression. Attention heads that exhibit only intra-image attention (top row) do not attend to other images and are not affected by the token compression performed within an image. Attention heads that use sinks for inter-image attention (lower row) use similar sinks for compressed tokens as well, demonstrating that the proposed delimiter token based attention sink approach is robust to pruning. To further demonstrate the potential of using BlindSight with token compression, we adapt DivPrune and measure the impact on the accuracy for Qwen2-VL (7B). Table 9 shows that combining BlindSight with 90% token compression still results in an accuracy within 2% of DivPrune.

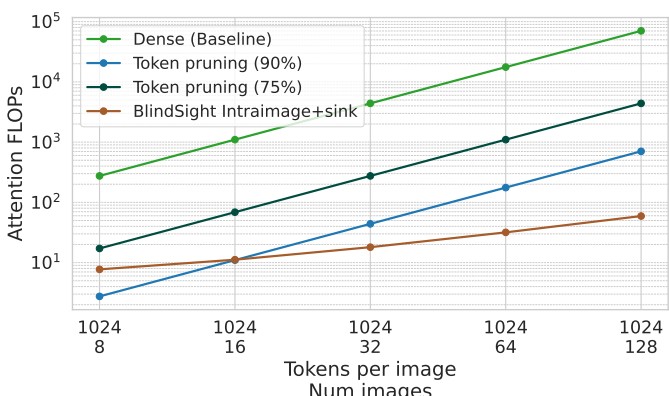

Figure 13: Theoretical attention FLOPs vs context length for BlindSight and token pruning.

| Model | MMIU | MANTIS | MuirBench |
|---|---|---|---|
| Baseline | 34.12 | 61.61 | 49.83 |
| BlindSight | 34.90 | 63.03 | 48.86 |
| DivPrune (90% token pruning) | 33.78 | 58.77 | 45.33 |
| BlindSight + DivPrune (90% token pruning) | 34.38 | 60.00 | 44.59 |
| DivPrune (75% token pruning) | 34.17 | 59.72 | 48.09 |
| BlindSight + DivPrune (75% token pruning) | 35.46 | 60.36 | 46.39 |

Table 9: Qwen2-VL (7B) performance for BlindSight with token compression

## K   SPARSE ATTENTION SPARSITY PATTERNS FOR VIDEO INPUTS

The Qwen3-VL model, unlike previous VLMs, introduces delimiter tokens that bound every frame in a video, along with text-based time stamps. This aligns videos to the multi-input image processing scheme leveraged by BlindSight. We observe a lack of inter-frame attention as a result. We aim to study extensions of BlindSight to video inputs in future work.

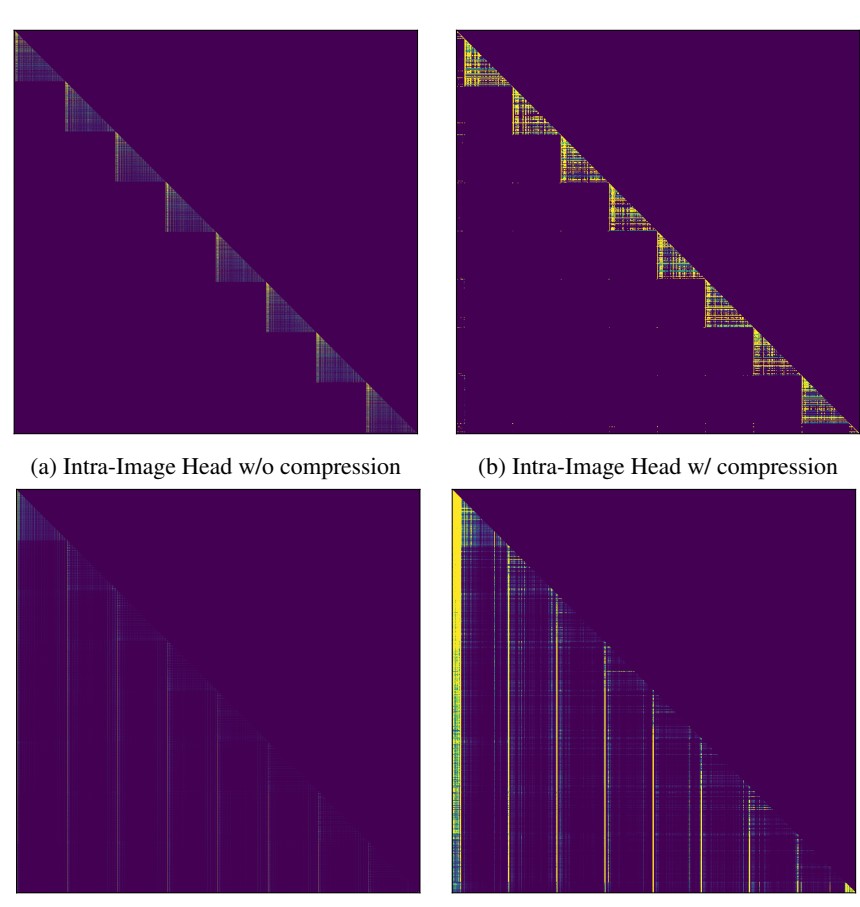

(a) Intra-Image Head w/o compression  (b) Intra-Image Head w/ compression

(c) Intra-Image+Sink Head w/o compression  (d) Intra-Image+Sink Head w/ compression

Figure 14: Attention scores w/ and w/o token compression. VLM sparsity patterns persist with $90\%$ token compression.

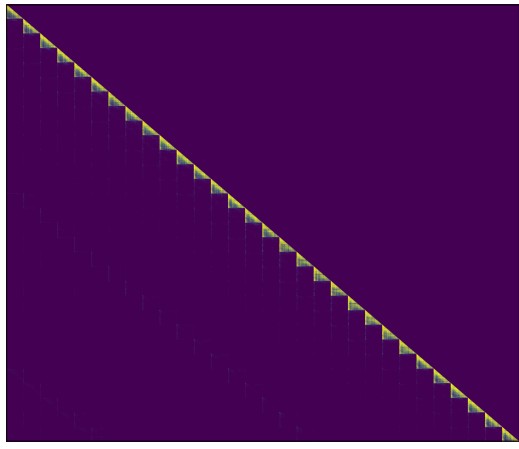

Figure 15: Sparse attention observed in Qwen3-VL (8B) processing videos.

## L BlindSight Attention Kernel

BlindSight assigns a sparsity pattern category to each attention head. The corresponding attention mask is constructed online for a given query using the indices of start and end tokens for text and images. To achieve practical speedup on GPUs, a custom compute kernel that leverages the available sparsity to reduce computation and runtime is required. This section describes the implementations and optimizations used in the BlindSight kernel.

### L.1 Overview

The proposed custom kernel is implemented using Triton. Similar to the Flash Attention kernel Dao et al. (2022), the BlindSight kernel divides the attention computation into smaller tiles. For sparse heads, the Triton-based kernel receives additional information alongside queries, keys, and values to avoid computing empty (i.e., completely sparse) tiles. We revert to the Triton-based Flash Attention kernel for dense heads.

The kernel consists of four different subroutines, where a suitable subroutine is selected for a given tile based on the types of query and key tokens (text or image) in the tile. Tiles handling text query tokens must visit all tiles in the key and value matrices and are therefore handled through a Triton-based flash attention subroutine (**dense_attention**). Tiles handling a mix of text and image query tokens will need to visit every tile in the key and value matrices. These may need to mask out inter-image or intra-image attentions depending on the head type. This case will be handled by a subroutine similar to the dense subroutine with additional masking logic to support the various head types. We refer to this as the **masked_attention** subroutine. Finally, tiles that handle only image query tokens can be completely skipped if all keys in a tile are masked out by the selected sparsity pattern. In addition to mask construction, logic is required to determine which tiles have unmasked keys that need to be processed. These are handled through two different subroutines; one to handle the attention sinks (**sink_attention**) and one to handle intra-image attention (**intra_image_attention**). We now discuss the subroutines and summarize optimizations.

### L.2 Notation

| Symbol | Space | Description |
|--------|-------|-------------|
| $B,L$ | $\mathbb{N}$ | Batch Size, Sequence Length |
| $H,D$ | $\mathbb{N}$ | Number of heads, Head dimension |
| $Q,K,V$ | $\mathbb{R}^{L \times D}$ | Query, key, and value matrices for a given batch and attention head*. |
| $m, n$ | $\mathbb{N}$ | Index along the query and key/value sequence lengths |
| $\text{block}_m, \text{block}_n$ | $\mathbb{N}$ | Tile size along the query and key/value sequence lengths. |
| $M$ | $\mathbb{N}$ | Total tiles in query sequence direction. $M = \lceil \frac{L}{\text{block}_m} \rceil$ |
| $N$ | $\mathbb{N}$ | Total tiles in key-value sequence direction. $N = \lceil \frac{L}{\text{block}_n} \rceil$ |
| $q$ | $\mathcal{R}^{\text{block}_m \times D}$ | Tile's query matrix in shared memory. |
| $k,v$ | $\mathcal{R}^{\text{block}_n \times D}$ | Tile's key and value matrices in shared memory. |

\* $Q, K$, and $V$ matrices are for a specific batch and head for brevity.

To construct the masks, the kernel requires extra information about each token. We first define an array imgid $\in \mathbb{N}^L$ for which an element is assigned a 0 if it is a text token, or an integer identifier of the image to which it belongs. From imgid, we can create a mask of text tokens istext $\in \mathbb{B}^L$ (True for all text positions) and a mask of image sinks isimgsink $\in \mathbb{B}^L$ (True for all sink positions). **sink_attention** and **intra_image_attention** require pointers to the location of the tiles with valid tokens (i.e., not completely sparse). $\text{sink}_{start} \in \mathbb{Z}$ points to the start of the image sinks and sink_end $\in \mathbb{Z}^M$ points to the end of the image sinks under the causal mask for each query tile. first_image $\in \mathbb{Z}^M$ points to the first location of the intra-image attention in a tile.

### L.3 Kernel Implementation

Algorithm 3 provides a top-level structure of the kernel and subroutine selection process. Figure 16 represents an example attention matrix, with the horizontal red lines representing the division of the

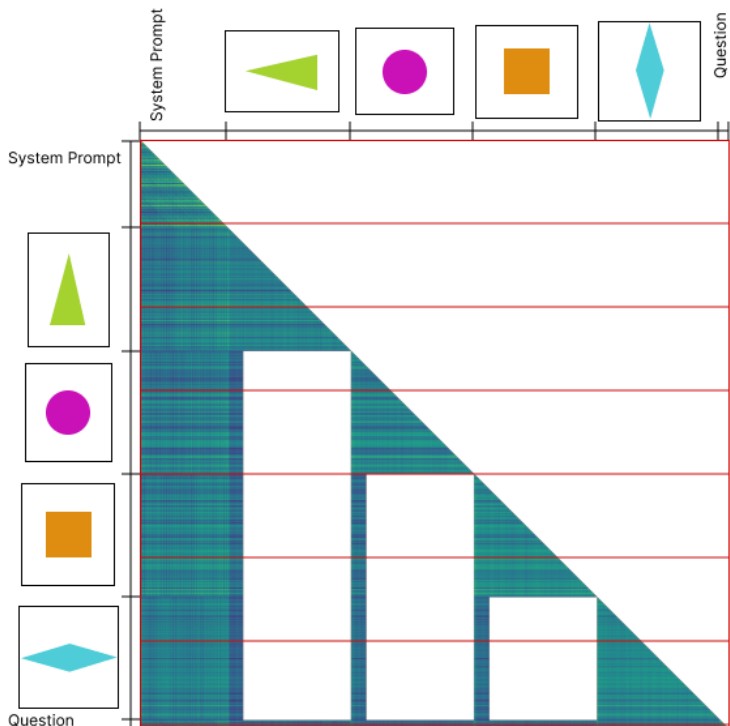

Figure 16: Intra-Image+Sink Head's attention matrix. Each red box outlines a row consisting all tiles for a subset of query tokens.

attention computation along the query dimension $(m)$. The kernel is parallelized along the batch, head, and query block dimensions, and $(B \times H \times \lceil \frac{L}{\text{block}_m} \rceil)$ workgroups/threadblocks are launched.

### L.3.1 DENSE ATTENTION SUBROUTINE

Algorithm 4 describes the subroutine applied to a row consisting of all text queries. In BlindSight, text queries follow the dense mask (with causal or sliding window as per the original model). The dense attention subroutine is applied to the first row in Fig. 16.

### L.3.2 MASKED ATTENTION SUBROUTINE

Algorithm 5 describes the attention subroutine applied to all tiles in a row consisting of image and text tokens. In Figure 16, the masked attention subroutine the last row.

### L.3.3 ATTENTION SINK SUBROUTINE

Algorithm 6 describes the subroutine applied to tiles that contain image-only queries and attention sinks. In BlindSight, an attention sink can appear due to text keys and inter-image attention sinks (highlighted by gold and indigo regions in Figure 17). A naïve approach would result in tiles that make poor use of sparsity considering that the image sinks are interspersed throughout the attention matrix (Fig. 18a).

To further optimize such a scenario, we propose reordering key and value matrices along the N dimension so that attention sinks are grouped together (Fig. 18b). As the kernel traverses the sequence of key/value tokens, the full tiles efficiently utilize the memory bandwidth and the GPU matrix-multiply units. To perform the permutation, we define the permutation vector $N^p \in \mathbb{N}^L$ as concat($\{i : \text{istext}_i = True\}, \{j : \text{isimgsink}_j = True\}, \{k : \neg\text{istext}_k \wedge \neg\text{isimgsink}_k\}$).

**Algorithm 3** BlindSight Kernel

---

**Input:**

$Q,K,V$                                                              (Query, key and value matrices)

$K^p\ V^p$                                                      (Permuted key and value matrices)

imgid                                                                 (imgid vector)

istext                                                                   (istext mask )

isimgsink                                                            (isimgsink mask)

$m$                                 (Start of the tile along the query sequence dimension $L$)

headtype                        (Mask type in use [dense,image,image_sink,sink])

$\text{sink}_{start}$           (The location of the first sink token in the key and value matrices.)

sink_end                      (Array of locations of the last sink in a query tile.)

first_image                    (Array of locations of the first image in a query tile.)

$\text{block}_m$                              (Size of the tile along the query matrix)

$\text{block}_n$                      (Size of the tile along the key and value matrices)

**Output** Out         (Attention output of tiles for a set of queries in a given attention head)

$q \leftarrow Q[m : m + \text{block}_m]$                                    $(q \in \mathcal{R}^{\text{block}_m \times D})$

$\text{istext}_m \leftarrow \text{istext}[m : m + \text{block}_m]$                    $(\text{istext}_m \in \mathbb{B}^{\text{block}_m})$

**if** $\text{all}(\text{istext}_m)$ **then**

    $\text{Out} \leftarrow \text{dense\_attention}(q, K, V, m, \text{block}_m, \text{block}_n)$              Alg. 4

**else if** $\text{any}(\text{istext}_m)$ **then**

    $\text{mask}_{kv} \leftarrow (\text{istext})$ if headtype = "image" otherwise $(\text{istext} \mid \text{isimgsink})$

    $\text{Out} \leftarrow \text{masked\_attention}(q, K, V, m, \text{istext}_m, \text{mask}_{kv}, \text{block}_m, \text{block}_n)$     Alg. 5

**else**

    $\text{mask}_{kv} \leftarrow (\text{istext})$ if headtype = "image" otherwise $(\text{istext} \mid \text{isimgsink})$

    $\text{sink}_{end} \leftarrow \text{sink\_end}[m]$

    $\text{Out} \leftarrow \text{sink\_attention}(q, K^p, V^p, m, \text{mask}_{kv}, \text{sink}_{start}, \text{sink}_{end}, \text{block}_m, \text{block}_n)$    Alg. 6

    **if** headtype = "image" or headtype = "image_sink" **then**

        $\text{Out} \leftarrow \text{intra\_image\_attention}(\text{Out}, q, K, V, \text{imgid}, \text{first\_image}, \text{block}_m, \text{block}_n)$   Alg. 7

    **end if**

**end if**

return Out

---

### L.3.4 INTRA-IMAGE ATTENTION SUBROUTINE

This subroutine is applied to tiles with only image tokens, i.e intra-image attention. In Fig. 17, the intra-image attention subroutine is applied to tiles containing green highlighted region. Algorithm 7 represents the approach for this subroutine. Note that since the intra-image attention is concentrated within a region, fewer tiles are required to compute this attention.

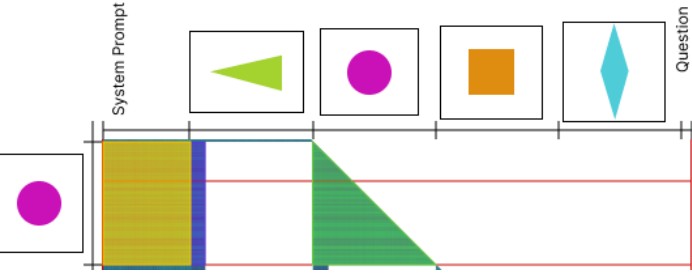

Figure 17: Two rows of the attention matrix shown in Fig. 16 containing only image query tokens. Segments of the attention scores are highlighted in gold, indigo, and green depending on whether they arise from image-text, inter-image, or intra-image attention respectively.

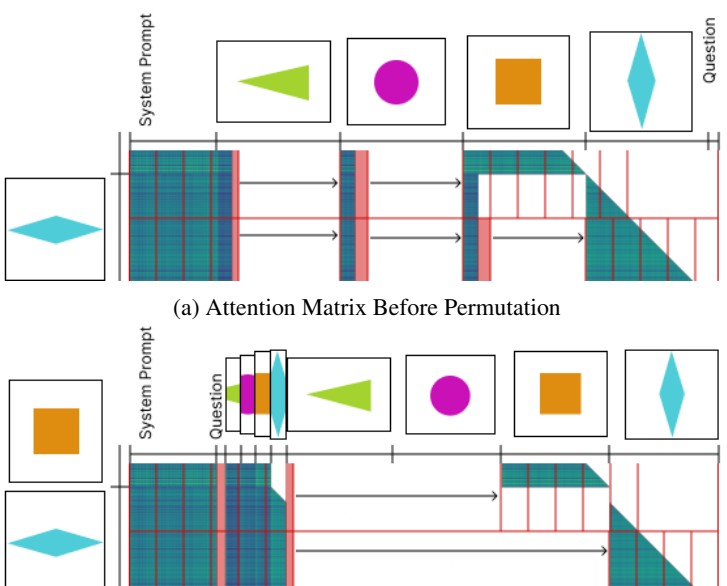

(a) Attention Matrix Before Permutation

(b) Attention Matrix After Permutation. Image sinks have been packed towards the front, saving the kernel from having to calculate an extra tile.

Figure 18: Two rows of the attention matrix shown in Fig. 16 which contain only image query tokens. Red lines outline tiles in both M and N dimensions. Areas highlighted in red show segments which fall within a tile but are numerically masked out, leading to poor GPU utilization.

---

**Algorithm 4** dense_attention

---

**Input:**
$q \in \mathcal{R}^{\text{block}_m \times D}$                                                         (Query tile)
$K, V$     (Key and value matrices)
$m$     (Start of the tile along the query sequence dimension)
$\text{block}_m$     (Size of the tile along the query sequence length)
$\text{block}_n$     (Size of the tile along the key/value sequence length)

**Output** $Out$

$m_{off} = \{m, m+1, m+2, ..., m+\text{block}_m - 1\}$     $(m_{off} \in \mathbb{N}^{\text{block}_m})$
$Out \leftarrow (0)_{\text{block}_m \times D}$     $(Out \in \mathcal{R}^{\text{block}_m \times D})$
$l \leftarrow (0)_{\text{block}_m}$     $(l \in \mathcal{R}^{\text{block}_m})$
$max \leftarrow (-\infty)_{\text{block}_m}$     $(max \in \mathcal{R}^{\text{block}_m})$
**for** $n$ in range$(0, m + \text{block}_m, \text{block}_n)$ **do**
    $k \leftarrow K[n : n + \text{block}_n]$     $(k \in \mathcal{R}^{\text{block}_n \times D})$
    $v \leftarrow V[n : n + \text{block}_n]$     $(v \in \mathcal{R}^{\text{block}_n \times D})$
    $s \leftarrow qk^T$
    $n_{off} = \{n, n+1, n+2, ..., n+\text{block}_n - 1\}$     $(n_{off} \in \mathbb{N}^{\text{block}_n})$
    $s'[i][j] \leftarrow s[i][j]$ if $i \geq j$ else $-\infty$ for $i \in m_{off}$ and $j \in n_{off}$
    $max_n \leftarrow \text{rowmax}(s')$
    $p \leftarrow \exp(s' - max_n)$
    $l_n \leftarrow \text{rowsum}(p)$
    $max^{new} \leftarrow \text{maximum}(max, max_n)$
    $l^{new} \leftarrow \exp(max - max^{new})l + \exp(max_n - max^{new})l_n$
    $Out \leftarrow \exp(max - max^{new})Out + \exp(max_n - max^{new})pv$
    $max \leftarrow max^{new}$
    $l \leftarrow l^{new}$
**end for**
return $Out$

---

---

**Algorithm 5** masked_attention

---

**Input:**
$q \in \mathcal{R}^{\text{block}_m \times D}$                                     (Query tile)
$K, V$                              (Key and value matrices)
$m$               (Start of the tile along the query matrices)
$\text{istext}_m \in \mathbb{B}^{\text{block}_m}$                            (Query mask)
$\text{mask}_{kv}$                     (Key, value matrix mask)
$\text{block}_m$              (Size of the tile along M dimension)
$\text{block}_n$              (Size of the tile along N dimension)

**Output** $Out$

$m_{off} = \{m, m+1, m+2, ..., m + \text{block}_m - 1\}$         $(m_{off} \in \mathbb{N}^{\text{block}_m})$
$Out \leftarrow (0)_{\text{block}_m \times D}$         $(Out \in \mathcal{R}^{\text{block}_m \times D})$
$l \leftarrow (0)_{\text{block}_m}$         $(l \in \mathcal{R}^{\text{block}_m})$
$max \leftarrow (-\infty)_{\text{block}_m}$         $(max \in \mathcal{R}^{\text{block}_m})$
**for** $n$ in range$(0, m + \text{block}_m, \text{block}_n)$ **do**
    $k \leftarrow K[n : n + \text{block}_n]$         $(k \in \mathcal{R}^{\text{block}_n \times D})$
    $v \leftarrow V[n : n + \text{block}_n]$         $(v \in \mathcal{R}^{\text{block}_n \times D})$
    $s \leftarrow qk^T$
    $mask_n \leftarrow \text{mask}_{kv}[n : n + \text{block}_n]$         $(mask_n \in \mathbb{B}^{\text{block}_n})$
    $mask \leftarrow \text{istext}_m[:, None] | mask_n[None, :]$     $(mask \in \mathbb{B}^{\text{block}_m \times \text{block}_n})$
    $n_{off} = \{n, n+1, n+2, ..., n + \text{block}_n - 1\}$     $(n_{off} \in \mathbb{N}^{\text{block}_n})$
    $mask_{causal}[i][j] \leftarrow 1$ if $i \geq j$ else $0$ for $i \in m_{off}$ and $j \in n_{off}$
    $s' \leftarrow s$ where $mask$ & $mask_{causal}$ otherwise $-\infty$
    $max_n \leftarrow \text{rowmax}(s')$
    $p \leftarrow \exp(s' - max_n)$
    $l_n \leftarrow \text{rowsum}(p)$
    $max^{new} \leftarrow \text{maximum}(max, max_n)$
    $l^{new} \leftarrow \exp(max - max^{new})l + \exp(max_n - max^{new})l_n$
    $Out \leftarrow \exp(max - max^{new})Out + \exp(max_n - max^{new})pv$
    $max \leftarrow max^{new}$
    $l \leftarrow l^{new}$
**end for**
return $Out$

---

---

**Algorithm 6** sink_attention

---

**Input:**

$q \in \mathcal{R}^{\text{block}_m \times D}$          (Query tile)

$K, V$          (Key and value matrices)

$N^p$          (Original index of each token)

$m$          (Start of the tile along the query matrices)

$\text{mask}_{kv}$          (Key, value token mask)

$\text{sink}_{start}$          (Start of each sink)

$\text{sink}_{end}$          (End of each sink)

$\text{block}_m$          (Size of the tile along M dimension)

$\text{block}_n$          (Size of the tile along N dimension)

**Output** $Out$

$m_{off} = \{m, m+1, m+2, ..., m+\text{block}_m - 1\}$      $(m_{off} \in \mathbb{N}^{\text{block}_m})$

$Out \leftarrow (0)_{\text{block}_m \times D}$      $(Out \in \mathcal{R}^{\text{block}_m \times D})$

$l \leftarrow (0)_{\text{block}_m}$      $(l \in \mathcal{R}^{\text{block}_m})$

$max \leftarrow (-\infty)_{\text{block}_m}$      $(max \in \mathcal{R}^{\text{block}_m})$

**for** $n$ in range($\text{sink}_{start}, \text{sink}_{end}, \text{block}_n$) **do**

     $n^p \leftarrow N^p[n : n + \text{block}_n]$      $(n \in \mathbb{N}^{\text{block}_n})$

     $k \leftarrow K^p[n : n + \text{block}_n]$      $(k \in \mathcal{R}^{\text{block}_n \times D})$

     $v \leftarrow V^p[n : n + \text{block}_n]$      $(v \in \mathcal{R}^{\text{block}_n \times D})$

     $s \leftarrow qk^T$

     $mask_n \leftarrow \text{mask}_{kv}[n^p]$      $(mask_n \in \mathbb{B}^{\text{block}_n})$

     $n_{off} = \{n, n+1, n+2, ..., n+\text{block}_n - 1\}$      $(n_{off} \in \mathbb{N}^{\text{block}_n})$

     $mask_{causal}[i][j] \leftarrow 1$ if $i \geq j$ else $0$ for $i \in m_{off}$ and $j \in n_{off}$

     $s' \leftarrow s$ where $mask_{causal}$ & $mask_n[None, :]$ otherwise $-\infty$

     $max_n \leftarrow \text{rowmax}(s')$

     $p \leftarrow \exp(s' - max_n)$

     $l_n \leftarrow \text{rowsum}(p)$

     $max^{new} \leftarrow \text{maximum}(max, max_n)$

     $l^{new} \leftarrow \exp(max - max^{new})l + \exp(max_n - max^{new})l_n$

     $Out \leftarrow \exp(max - max^{new})Out + \exp(max_n - max^{new})pv$

     $max \leftarrow max^{new}$

     $l \leftarrow l^{new}$

**end for**

return $Out$

---

---

**Algorithm 7** intraimage_attention

**Input:**

$Out \in \mathcal{R}^{\text{block}_m \times D}$        (Previous $Out$ value for this subroutine to accumulate)
$q \in \mathcal{R}^{\text{block}_m \times D}$        (Query tile)
$K,V$        (Key and value matrices)
$m$        (Start of the tile along the query matrix)
imgid        (The image id vector)
first_image        (Location of the first image)
$\text{block}_m$        (Size of the tile along M dimension)
$\text{block}_n$        (Size of the tile along N dimension)

**Output** $Out$

$m_{off} = \{m, m+1, m+2, ..., m+\text{block}_m - 1\}$        $(m_{off} \in \mathbb{N}^{\text{block}_m})$
$l \leftarrow (0)_{\text{block}_m}$        $(l \in \mathcal{R}^{\text{block}_m})$
$max \leftarrow (-\infty)_{\text{block}_m}$        $(max \in \mathcal{R}^{\text{block}_m})$
$\text{first}_{image} = \text{first\_image}[m \div \text{block}_m]$
$\text{imgid}_m \leftarrow \text{imgid}[m : m+\text{block}_m]$        $(\text{imgid}_m \in \mathbb{N}^{\text{block}_m})$
**for** $n$ in range($\text{first}_{image}, m+\text{block}_m, \text{block}_n$) **do**
    $k \leftarrow K[n : n+\text{block}_n]$        $(k \in \mathcal{R}^{\text{block}_n \times D})$
    $v \leftarrow V[n : n+\text{block}_n]$        $(v \in \mathcal{R}^{\text{block}_n \times D})$
    $\text{imgid}_n \leftarrow \text{imgid}[n : n+\text{block}_n]$        $(\text{imgid}_n \in \mathbb{N}^{\text{block}_n})$
    $s \leftarrow qk^T$
    $mask \leftarrow \text{imgid}_m[:, None] = \text{imgid}_n[None, :]$        $(mask \in \mathbb{B}^{\text{block}_m \times \text{block}_n})$
    $n_{off} = \{n, n+1, n+2, ..., n+\text{block}_n - 1\}$        $(n_{off} \in \mathbb{N}^{\text{block}_n})$
    $mask_{causal}[i][j] \leftarrow 1$ if $i \geq j$ else $0$ for $i \in m_{off}$ and $j \in n_{off}$
    $s' \leftarrow s$ where $mask \ \& \ mask_{causal}$ otherwise $-\infty$
    $max_n \leftarrow \text{rowmax}(s')$
    $p \leftarrow \exp(s' - max_n)$
    $l_n \leftarrow \text{rowsum}(p)$
    $max^{new} \leftarrow \text{maximum}(max, max_n)$
    $l^{new} \leftarrow \exp(max - max^{new})l + \exp(max_n - max^{new})l_n$
    $Out \leftarrow \exp(max - max^{new})Out + \exp(max_n - max^{new})pv$
    $max \leftarrow max^{new}$
    $l \leftarrow l^{new}$
**end for**
return $Out$

---

## M    ETHICS STATEMENT

We adhered to the principles outlined in the ICLR Code of Ethics throughout this research. To ensure scientific integrity, we have made our implementations publicly available for reproducing all the results. AI-based tools were used to polish the text. We have also made efforts to acknowledge all relevant prior publications.

## N    REPRODUCIBILITY STATEMENT

The codebase for reproducing all the results is available at `https://anonymous.4open.science/r/BlindSight-AFDC`. The approach can be easily extended to the current and future VLMs released on Hugging Face.

