# OpenReview forum: "BlindSight: Harnessing Sparsity for Efficient Vision-Language Models"
_ICLR.cc/2026/Conference — Submitted to ICLR 2026_

### Official Review · Reviewer_T2Dp · 2025-10-26

**Soundness:** 3
**Presentation:** 3
**Contribution:** 3
**Rating:** 6
**Confidence:** 5

**Summary:**

The paper’s motivation is to enable efficient prefilling (reduce time to first token) for vision-language models that have several visual tokens in their context corresponding to many images. The paper’s approach is to study attention patterns, particularly inter-image attention patterns. They categorize attention patterns into four categories: dense, sink, intra-image, and sink + intra-image. This is done empirically by visualizing the attention matrix for some trained VLMs on a dataset. The paper then proposes BlindSIGHT, a post-training algorithm that limits each attention head to one of the four templates. For a given model, this is done through calibration stages, where the model is run on a dataset (the paper used MMIU) to identify the dominant sparsity pattern of each attention head and assign it to one of the four templates. The paper shows that a Triton implementation of their post-training method results in more than a 2× improvement in TTFT when the context length is as long as 300k tokens, with about a 1% accuracy loss across different multi-frame VLM benchmarks. Finally, the paper provides intuition and empirical evidence on when attention sinks occur in VLMs, arguing that they mainly appear for image boundary delimiter tokens.

**Strengths:**

- The research direction pursued in the paper (addressing the efficiency challenge of VLMs when processing multiple images due to long context) is important and has clear real-world benefits.
- The paper’s presentation is easy to follow.
- The analysis of sparsity patterns in VLM heads for multi-image input is interesting and valuable for future research in the field.
- The proposed method (BlindSIGHT) is simple yet effective: it converts as many attention heads as possible to more efficient sparsity patterns. In addition, the Triton-based implementation shows significant speedup, achieving more than a 2× reduction in TTFT. Although the proposed method is post-hoc, the paper motivates future work to incorporate such sparse templates during training, which is a logical next step.

**Weaknesses:**

- One major issue with the proposed method is the need to adjust several hyper-parameters: $\alpha_{layer}$, $\gamma_d$, $\gamma_s$, $\gamma_i$, and the choice of 10% for the sink sparsity template. The paper suggests finding reasonable values through a “calibration” process on a given model and dataset (e.g., MMIU). However, it remains unclear to what extent these optimal values transfer to queries or domains that differ significantly from the calibration dataset in real world use cases.
- The paper mentions that the degree of enforced sparsity (through the chosen hyper-parameters) determines the trade-off with accuracy. For the settings reported, an average accuracy drop of about 1% is observed. However, it remains unclear how this trade-off behaves more broadly. The authors could provide an accuracy–TTFT plot showing how performance changes as the hyper-parameters vary.
- A key contribution of the paper is replacing dense inter-image attention with sparse templates. However, there is no discussion on why inter-image attention could be useful. What types of queries require inter-image attention? In Section 6.2, results on various multi-image benchmarks are presented, but the paper should discuss what each evaluation measures, what aspects might require inter-image attention, and how the proposed sparse templates handle such cases.
- The paper mainly presents results for QwenVL and Gemma. It is unclear whether the observed sparsity patterns would remain relevant for other VLMs, especially those designed to produce fewer visual tokens (for example, FastVLM [1] that generates 16× fewer visual tokens than regular ViTs).

[1] Vasu, Pavan Kumar Anasosalu, et al. “FastVLM: Efficient Vision Encoding for Vision Language Models.” Proceedings of the Computer Vision and Pattern Recognition Conference. 2025.

**Questions:**

- In line 357 (Infrastructure), which VLM are you referring to?
- For the results shown in Fig. 4, what version of FlashAttention (FA) was used as the baseline? Would you still observe improvements when using the proposed Triton-based kernel compared to more recent variants of FA?

---

> ### Author Response · Authors · 2025-11-23
> **Author Response**
>
> Thank you for a thorough review of the paper! We appreciate your feedback and questions. We are glad to see your recognition of the technique's potential impact on designing efficient VLM architectures. We hope to have addressed your comments in our updated manuscript.
>
> > **One major issue with the proposed method is the need to adjust several hyper-parameters: , , , , and the choice of 10% for the sink sparsity template. The paper suggests finding reasonable values through a “calibration” process on a given model and dataset (e.g., MMIU). However, it remains unclear to what extent these optimal values transfer to queries or domains that differ significantly from the calibration dataset in real world use cases.**
>
> We recommend that users by default set $\alpha_{layer}=0.1$, $\gamma_d=0.25$, and $\gamma_s=\gamma_i=0.6$, values, which perform consistently across typical VLM architectures through our experimentation. We tested these parameters on the latest Qwen3-VL model and achieved consistent performance out of the box.
>
> | Qwen3 VL (32B) | MMIU | MANTIS | MuirBench |
> |---------------|-------|--------|-----------|
> | Baseline | 48.35 | 79.15 | 70.98 |
> | BlindSight | 46.56 | 78.20 | 70.55 |
>
> Among all hyperparameters, $\alpha_{layer}$ emerges as the key hyperparameter influencing performance. We observe that accuracy exhibits a clear and predictable trend with respect to $\alpha$. $\gamma_d$ is also a hyperparameter that affects the number of dense heads and accuracy if changed significantly (e.g., from $0.1$ to $0.4$ in Table 3, Appendix D). The clear trend between hyperparameter values and accuracy allows users to leverage standard hyperparameter tuning techniques to identify the optimal value efficiently.
>
> We would also like to highlight that the multi-image comprehension benchmarks evaluated cover a diverse range of scenarios, as discussed in Section 6.2. We also experimented with an alternative dataset for characterization (Appendix F), where performance remained consistent with the original MMIU characterization. Although performance losses may occur in real-world scenarios, we hypothesize that fine-tuning with sparsity masks can mitigate these losses. In the future, we hope that efficient VLMs incorporating BlindSight-inspired sparsity patterns will be designed during the pre-training stage.
>
> **Changes in the revised paper:** Expanded Appendix D with more hyperparameter ablation studies, added Appendix E to highlight the variety of tasks performed by evaluated datasets, and added Appendix F to show the robustness of the default suggested hyperparameters across characterization datasets.
>
>
> > **The paper mentions that the degree of enforced sparsity (through the chosen hyper-parameters) determines the trade-off with accuracy. For the settings reported, an average accuracy drop of about 1% is observed. However, it remains unclear how this trade-off behaves more broadly. The authors could provide an accuracy–TTFT plot showing how performance changes as the hyper-parameters vary.**
>
> We have now included statistics covering the FLOPs savings in the attention layer per benchmark in Appendix H to complement the accuracy results presented in Table 1. We observe a median savings in the range of $29\%-48\%$ for the Qwen models and $12\%$-$46\%$ for Gemma models. Additionally, we have included an analysis comparing the trade-off between accuracy and FLOPs saved in the attention computation across different sparsity levels in Appendix I. We believe that users can leverage this analysis to identify the knee point where accuracy saturates.
>
> **Changes in the revised version:** We have added Appendix H and I to include compute-accuracy trade-offs.

---

> ### Author Response · Authors · 2025-11-23
> **Author Response**
>
> > **A key contribution of the paper is replacing dense inter-image attention with sparse templates. However, there is no discussion on why inter-image attention could be useful. What types of queries require inter-image attention? In Section 6.2, results on various multi-image benchmarks are presented, but the paper should discuss what each evaluation measures, what aspects might require inter-image attention, and how the proposed sparse templates handle such cases.**
>
> Inter-image attention is most relevant for tasks requiring reasoning across multiple images, such as Ordering or temporal reasoning (e.g., “Describe the action sequence across these frames”), Similarity and retrieval (e.g., “Find the image most similar to the first”), and Cross-image comparison (e.g., “Identify objects that appear in both images”). Per your recommendation, we have also added a section (Appendix E) to provide further details on each benchmark evaluated. Benchmarks like MuirBench include such tasks, and we observe that replacing all heads with intra-image masks (no inter-image attention) significantly reduces accuracy (e.g., Qwen2.5-VL (32B): $61.87\%$ → $49.14\%$). BlindSight mitigates this by performing offline characterization and preserving attention heads that perform inter-image attention through dense attention or attention sinks.
>
> **Changes in the revised paper:** Expanded Table 1 to include accuracy results for fixed intra-image mask (no inter-image attention) to highlight the importance of inter-image attention.
>
> > **The paper mainly presents results for QwenVL and Gemma. It is unclear whether the observed sparsity patterns would remain relevant for other VLMs, especially those designed to produce fewer visual tokens (for example, FastVLM [1] that generates 16× fewer visual tokens than regular ViTs).**
> > [1] Vasu, Pavan Kumar Anasosalu, et al. “FastVLM: Efficient Vision Encoding for Vision Language Models.” Proceedings of the Computer Vision and Pattern Recognition Conference. 2025.
>
> We believe that multi-image sparsity patterns for typical VLMs (causal intra-image attention) will follow patterns showcased for the Qwen models. We have added sparsity patterns for Llama 4 Scout (17B) and Qwen3-VL (8B) in Appendix A. We would like to highlight that the Gemma models evaluated tokenize the image into 128 tokens each, though their design uses a non-causal intra-image attention, which is very different from the Qwen architecture. While we did attempt alternative models (FastVLM, GLM-4.5V), these models don't have APIs present to access attention scores within the models yet. We hope to validate these observations once multi-image processing features become available in other models.
>
> Furthermore, we have observed similar sparsity patterns for different image resolutions and with token compression approaches (e.g., DivPrune with $90\%$ pruning) that produce significantly fewer visual tokens (Appendix J), which strengthens our proposed sparsity pattern's applicability.
>
> **Changes in the revised paper:** Added Appendices A and J to include attention sparsity patterns for new models and token compression approaches.
>
> > **In line 357 (Infrastructure), which VLM are you referring to?**
>
> We have removed this sentence to prevent confusion. This statement applies only when running inference without BlindSight’s sparse kernel in eager mode. We are able to evaluate every benchmark in this paper on a single GPU.
>
> > **For the results shown in Fig. 4, what version of FlashAttention (FA) was used as the baseline? Would you still observe improvements when using the proposed Triton-based kernel compared to more recent variants of FA?**
>
> The baseline results shown in Fig. 4 use a Triton implementation of FlashAttention v2. The sparse kernel is similarly implemented in Triton and based on FlashAttention v2. When considering newer versions of FlashAttention or alternate implementations (Composable Kernel, etc.), it is important to note that BlindSight improves the speed of the attention kernel by reducing the number of tiles the kernel must compute. BlindSight is orthogonal to the tile-level attention computation or parallelization across tiles, meaning it could leverage the same techniques as FlashAttention v3 and FlashAttention v4. For example, FlashAttention v3 introduces hardware-accelerated low-precision GEMMs using FP8 Tensor Cores. Such a technique could easily be adapted to BlindSight. Moreover, FlashAttention v3's usage of asynchronous WGMMA operations to hide the latency of low-throughput operations, such as floating-point exponential, could also be used to hide the latency of BlindSight's mask calculations. This implies that Blindsight may achieve an even greater speedup when adapted to more recent variants of FlashAttention.
>
> Thank you again for taking the time to provide constructive feedback to us! We hope that our latest manuscript has increased your belief in our work.

---

### Official Review · Reviewer_7HSk · 2025-10-27

**Soundness:** 2
**Presentation:** 2
**Contribution:** 2
**Rating:** 2
**Confidence:** 4

**Summary:**

This work introduces a training-free framework to accelerate VLMs under multi-image inference scenarios. It presents a prompt-template-aware sparsity mask derived from analyzing attention maps, categorizing attention heads into four structural types: Dense, Sink, Intra-Image, and Intra-Image+Sink. The method further employs a dataset-level aggregation scheme to convert prompt-dependent sparsity patterns into a prompt-agnostic configuration. Experiments were conducted on Qwen2-VL, Qwen2.5-VL, and Gemma 3 models.

**Strengths:**

- This study empirically characterizes four recurring head-level attention patterns across major VLM families such as Qwen and Gemma, providing insights into modality-aware sparsity in multimodal transformers.
- This study presents a Triton-based attention kernel tailored for the proposed method, achieving performance gains in realistic long-context inference tasks.

**Weaknesses:**

- I think this work should be compared with previous vision token pruning and token merging methods for VLMs, such as FastV, LLaVA-PruMerge, DivPrune, and DART, in terms of reducing the computational cost of vision tokens. Currently, only a comparison with the original model as the baseline is provided, and the experimental section therefore feels relatively weak.
  * [FastV] https://arxiv.org/abs/2403.06764
  * [LLaVA-PruMerge] https://arxiv.org/abs/2403.15388
  * [DivPrune] https://arxiv.org/abs/2503.02175
  * [DART] https://arxiv.org/abs/2502.11494

- The choice of algorithm parameters (alpha_layer, gamma_d, gamma_s, gamma_i) seems largely heuristic. Providing rationale, sensitivity analysis, or a principled selection rule for these values would improve the reproducibility and robustness of the approach.

- The accuracy results in Table 1 are reported without specifying the corresponding sparsity levels or inference gains, making it hard to interpret the trade-offs.

- The prompt-level characterization seems to rely primarily on the MMIU benchmark. It remains unclear how well the identified sparsity patterns generalize to other datasets. An analysis of cross-dataset generalizability would be valuable.

- The prompt-level characterization process does not seem computationally lightweight. It would be helpful if the authors could provide information about the associated computational time.

- The analysis presented in this work is interesting; however, it remains unclear whether the proposed sparsity characterization generalizes to temporally correlated multi-image settings such as video frames. In my experience, video frames can also be treated as separate images with distinct delimiter tokens, often yielding comparable performance in video-based VLM inference. In such cases, certain layers are likely to exhibit inter-frame cross-attention patterns due to the high similarity among visual features. It would be valuable to analyze these correlations and discuss how the proposed method behaves when applied to video-like inputs.

**Questions:**

Please find the Weakness section.

---

> ### Author Response · Authors · 2025-11-23
> **Author Response**
>
> Thank you for your detailed review of our paper! We have updated our manuscript based on your constructive feedback.
>
> >  **I think this work should be compared with previous vision token pruning and token merging methods for VLMs, such as FastV, LLaVA-PruMerge, DivPrune, and DART, in terms of reducing the computational cost of vision tokens. Currently, only a comparison with the original model as the baseline is provided, and the experimental section therefore feels relatively weak.**
> > 1. [FastV]  [https://arxiv.org/abs/2403.06764](https://arxiv.org/abs/2403.06764)
> > 2. [LLaVA-PruMerge]  [https://arxiv.org/abs/2403.15388](https://arxiv.org/abs/2403.15388)
> > 3. [DivPrune]  [https://arxiv.org/abs/2503.02175](https://arxiv.org/abs/2503.02175)
> > 4. [DART]  [https://arxiv.org/abs/2502.11494](https://arxiv.org/abs/2502.11494)
>
> We appreciate the suggestion and highlight that token pruning/merging methods (FastV, LLaVA-PruMerge, DivPrune, DART) address efficiency differently. There are several works that have focused on different directions for reducing the inference and training runtime, including token pruning/merging, model compression (quantization), and sparse attention. In this work, we focus on sparse attention direction.
>
> **Key distinction:**
>
> - **No runtime metrics:** Several token compression approaches perform runtime characterization to prune or merge tokens using attention scores. In this work, we focus on an offline approach to eliminate the runtime overhead of any online characterization.
>
> - **Focus on multi-image inputs:** Existing token pruning and merging approaches have focused on single-image or video benchmarks. In contrast, BlindSight focuses on multi-image vision inputs. The lack of evaluation on multi-image benchmarks makes it difficult to perform a meaningful quantitative comparison between BlindSight and existing token compression approaches.
>
> - **Sub-Quadratic scaling for BlindSight's sparse attention**: Token compression approaches are efficient, however, they might result in a quadratic increase in attention computation if fixed compression rate is used. The proposed attention sparsity patterns in BlindSight provide sub-quadratic attention FLOPs with sequence length, making it a more comput- efficient option for a higher number of images in VLM. Figure 13 in the revised paper compares theoretical attention FLOPs for Dense attention, token compression (fixed compression rate of 90% and 75%), and BlindSight Intra-image+Sink attention. A the number of images increases, BlindSight sparse attention results in lower FLOPs due to sub-quadratic increase in compute.
>
> - **Orthogonal to token compression**: Since we focus on inter-image attention and several token compression approaches focus on reducing the number of tokens per image, we believe our work is orthogonal to these approaches and can be combined for further improvement.
>
> 	To demonstrate the potential of combining BlindSight with token compression, we adapt DivPrune for multi-image inputs. We apply the token compression strategy proposed by DivPrune to each image and analyze the attention sparsity patterns for the token compression scenario in Figure 14, which shows that the attention sparsity patterns observed in the baseline models are retained with compressed tokens ($90$\% token compression) as well. Accuracy results in Table 9 also validate this observation, which shows that **BlindSight can be used with DivPrune** ($90$\% token compression) and achieves $-0.8$\% to $+1.2$\% accuracy compared to only DivPrune ($90\%$ token compression).
>
> |Qwen2 VL (7B)   | MMIU | MANTIS | MUIRBench |
> |--|--|--|--|
> | Baseline | 34.12 | 61.61 | 49.83 |
> | BlindSight | 34.90 | 63.03 | 48.86 |
> | DivPrune (90% token pruning) | 33.78 | 58.77 | 45.33 |
> | BlindSight + DivPrune (90% token pruning) | 34.38 | 60.00 | 44.59 |
>
>
> Attention heads that exhibit only intra-image attention do not attend to other images and are not affected by token compression performed within an image. Attention heads that use sinks for inter-image attention apply similar sinks to compressed tokens as well, demonstrating that the proposed delimiter-token-based attention sink approach is robust to different tokenization methods and optimizations.
>
> **Changes in the revised paper**: We have added Appendix J for qualitative and quantitative comparison with token compression.

---

> ### Author Response · Authors · 2025-11-23
> **Author Response**
>
> > **The choice of algorithm parameters (alpha_layer, gamma_d, gamma_s, gamma_i) seems largely heuristic. Providing rationale, sensitivity analysis, or a principled selection rule for these values would improve the reproducibility and robustness of the approach.**
>
> BlindSight's hyperparameters are designed for lightweight tuning for a domain if required. As with any practical engineering solution, certain design parameters require calibration based on the target application. To address this concern, we have updated the manuscript to clarify that, through extensive experimentation, we identified a parameter set that minimizes performance degradation: $\alpha_{layer}=0.1$, $\gamma_d=0.25$, and $\gamma_s=\gamma_i=0.6$. For typical VLMs following Qwen/Llama-style architectures, we recommend users conduct a lightweight experiment on their end use case to balance accuracy and FLOPs reduction, selecting the knee point for  $\alpha_{layer}$​ below which accuracy saturates (Appendix I). Notably, the parameters  $\gamma_d$, $\gamma_s$, $\gamma_i$  generally do not require tuning.
>
> Among all hyperparameters, $\alpha_{layer}$ emerges as the key hyperparameter influencing performance. We observe that accuracy exhibits a clear and predictable trend with respect to $\alpha$. $\gamma_d$ is also a hyperparameter that affects the number of dense heads and accuracy if changed significantly (e.g., from $0.1$ to $0.4$ in Table 3, Appendix D). We observed a clear trend between hyperparameter values and accuracy, which allows users to leverage standard hyperparameter tuning techniques to identify the optimal value efficiently.
>
> Furthermore, we demonstrate that once selected for a model, the same hyperparameter values can be applied across different model sizes within the same family and across multiple datasets. When experimenting on a new Qwen3-VL model, we observed performance trends similar to those in Table 1.
>
> | Qwen3 VL (32B) | MMIU | MANTIS | MuirBench |
> |---------------|-------|--------|-----------|
> | Baseline | 48.35 | 79.15 | 70.98 |
> | BlindSight | 46.56 | 78.20 | 70.55 |
>
> **Changes in the paper:** Expanded Appendix D for ablation studies on different hyperparameters.
>
> > **The accuracy results in Table 1 are reported without specifying the corresponding sparsity levels or inference gains, making it hard to interpret the trade-offs.**
>
> We have updated the manuscript to include attention FLOPs reduction statistics (Appendix H) and a detailed accuracy–efficiency trade-off analysis (Appendix I). Appendix I shows a clear knee point for $\alpha_{layer}$ below $0.1$, accuracy saturates while FLOPs savings diminish, enabling principled tuning. These additions clarify the trade-offs between sparsity, accuracy, and inference gains.
>
> **Changes in the revised paper:** Added Appendix H and I to include FLOPs reduction and sparsity-accuracy tradeoff.
>
> > **The prompt-level characterization seems to rely primarily on the MMIU benchmark. It remains unclear how well the identified sparsity patterns generalize to other datasets. An analysis of cross-dataset generalizability would be valuable.**
>
> We further studied the robustness of BlindSight with alternative characterization datasets. With this in mind, we have included an experimental study in Appendix F, where we showcase the  robustness of BlindSight to the characterization dataset. For Qwen2.5-VL (32B), accuracy on MuirBench is $61.64\%$ when characterized on MMIU vs $62.06\%$ when characterized on MuirBench. We recommend selecting characterization datasets with varied image numbers and task types for the best coverage.
>
> **Changes in the revised paper:**  Added Appendix F
>
> > **The prompt-level characterization process does not seem computationally lightweight. It would be helpful if the authors could provide information about the associated computational time.**
>
> The prompt-level characterization process is essential to identify the optimal combination of dense and sparse attention heads in the VLM. Given the variation in the attention head types across different prompt structures in a dataset, the aggregation scheme aims to mitigate performance losses. The execution time for running a characterization is equivalent to the time required for running the benchmark itself in eager mode (access to attention scores required), along with the overhead of computing the attention scores for different masking strategies. For example, on Qwen2.5-VL (7B), characterization on the MMIU dataset ($\sim2,000$ prompts) required less than $1$ GPU-hour, which is negligible compared to training or fine-tuning. We would like to highlight that a new model only requires a single characterization run to derive the attention head to sparsity type mapping,  which can be used for all dataset evaluations (Table 1).

---

> ### Author Response · Authors · 2025-11-23
> **Author Response**
>
> >  **The analysis presented in this work is interesting; however, it remains unclear whether the proposed sparsity characterization generalizes to temporally correlated multi-image settings such as video frames. In my experience, video frames can also be treated as separate images with distinct delimiter tokens, often yielding comparable performance in video-based VLM inference. In such cases, certain layers are likely to exhibit inter-frame cross-attention patterns due to the high similarity among visual features. It would be valuable to analyze these correlations and discuss how the proposed method behaves when applied to video-like inputs.**
>
> Thank you for your comment. One of our core objectives, while highlighting the presence of sparse layers with repeated patterns in VLMs, is to encourage model developers to incorporate BlindSight-like sparsity into the model architecture itself. Historically, a key limitation for video inputs was the absence of per-frame delimiter tokens. As discussed in Section 7, these tokens play a key role in enabling sparsity. We are encouraged by recent developments, such as Qwen3-VL, which has introduced delimiter tokens per frame, along with timestamps. With this, we observe the presence of BlindSight-like sparsity patterns even in videos (Appendix K). We anticipate expanding the proposed sparsity pattern classes to include temporal correlation (diagonal lines in the attention heatmap) between frames. We plan to extend BlindSight to fully support video inputs in future work.
>
> **Changes in the revised paper:** Added Appendix K to include attention sparsity pattern for video inputs in Qwen3 VL.
>
> Thank you for your valuable feedback and constructive critique! We have focused on incorporating additional studies from your comments into our latest manuscript. We hope this strengthens your confidence in BlindSight.

---

> > ### Comment · Reviewer_7HSk · 2025-11-26
> >
> > I appreciate the authors’ response and the additional experiments. The comparison with DivPrune and the combination of BlindSight+DivPrune are particularly interesting. It would be helpful to also include results for DivPrune at 75%, not only at 90%.
> >
> > Furthermore, the ablation studies on hyperparameter choices, characterization datasets, and experiments with additional models strengthen the empirical contributions of this work. I would like to increase my score from 2 to 4. I also hope the authors can properly fill in the remaining * entries in Table 1.
> >
> > Most of the benchmarks used in the paper seem to involve multiple-choice questions, where the model only needs to generate short answers such as option A or B. Reporting results on generative examples (e.g., descriptive answers about the images or other open-ended questions) would further improve the convincingness of this work.

---

> > > ### Author Response · Authors · 2025-12-03
> > > **Author response**
> > >
> > > Thank you for looking into our additional experimental results. We appreciate your increased confidence in our work and your constructive feedback! We have added further details and additional results based on your feedback. We have also updated the manuscript with all results.
> > >
> > > > **Divprune+BlindSight evaluation for 75% token compression:**
> > >
> > > Based on your request, we conducted additional experiments with 75% pruning. These results indicate that BlindSight maintains both compatibility and efficiency across different token compression rates. We emphasize that such VLM optimization techniques are orthogonal to BlindSight, which addresses an inherent inefficiency in current VLM architectures.
> > >
> > > |Qwen2 VL (7B)  | MMIU | MANTIS | MUIRBench |
> > > |--|--|--|--|
> > > | Baseline | 34.12 | 61.61 | 49.83 |
> > > | BlindSight | 34.90 | 63.03 | 48.86 |
> > > | DivPrune (90% token pruning) | 33.78 | 58.77 | 45.33 |
> > > | BlindSight + DivPrune (90% token pruning) | 34.38 | 60.00 | 44.59 |
> > > | DivPrune (75% token pruning) | 34.17 | 59.72 |  48.09|
> > > | BlindSight + DivPrune (75% token pruning) | 35.46 | 60.36 | 46.39  |

---

> > > ### Author Response · Authors · 2025-12-03
> > > **Author response**
> > >
> > > > **BlindSight for generative examples:**
> > >
> > > We agree that most benchmarks evaluated in this work involve multiple-choice questions or single-word answers. However,  while these questions in the benchmarks only require single letter responses, recent VLMs by default generate a sequence of text for reasoning. We have included a few samples from benchmarks on the Qwen2.5 32B model below for reference, where the selected answer is at the end. This is a common feature across all the benchmarks evaluated.
> > >
> > > *Example 1:*
> > > ```
> > > Question: <image1><image2>If the ABCD shape is a rectangle, the perimeter of the ABCD rectangle is equal to the perimeter of the EFGH parallelogram, compute the length of the EH side of the EFGH parallelogram. Round computations to 2 decimal places.
> > >
> > > Ground truth answer: 9.0
> > >
> > > BlindSight Answer:
> > >
> > > Here's how we can solve this problem, combining both a human-like explanation and a structured breakdown:
> > >
> > > **Understanding the Problem**
> > >
> > > We're given that ABCD is a rectangle, and EFGH is a parallelogram. We know the perimeter of the rectangle ABCD is equal to the perimeter of the parallelogram EFGH. We need to find the length of the EH side of the parallelogram. To do this, we need to compute the perimeter of the rectangle ABCD using the given information from the diagram. Once we have that perimeter, we can use it to find the lengths of the sides of the parallelogram EFGH and solve for EH.
> > >
> > > **Method 1: Human Explanation**
> > >
> > > 1.  **Find the perimeter of rectangle ABCD:** The diagram tells us that AB = 7 and BC = 12. The perimeter of a rectangle is 2 * (length + width).  So, the perimeter of ABCD is 2 * (7 + 12) = 2 * 19 = 38.
> > > 2.  **Use the perimeter to find the sides of parallelogram EFGH:** Since the perimeters are equal, the perimeter of EFGH is also 38. A parallelogram has two pairs of equal sides. Let EF = HG and EH = FG.  The perimeter of a parallelogram is 2 * (side1 + side2). So, 2 * (EF + EH) = 38. This simplifies to EF + EH = 19.
> > > 3.  **Use the given information about EF:** The diagram tells us that EF = 10.
> > > 4.  **Solve for EH:** Substitute EF = 10 into the equation EF + EH = 19.  This gives us 10 + EH = 19.  Therefore, EH = 19 - 10 = 9.
> > >
> > > **Method 2: GPT Breakdown**
> > > 1.  **Rectangle ABCD Perimeter:**\n  *  AB = 7\n  *  BC = 12
> > >   *  Perimeter (ABCD) = 2 * (AB + BC) = 2 * (7 + 12) = 2 * 19 = 38
> > > 2.  **Parallelogram EFGH Perimeter:**
> > >   *  Perimeter (EFGH) = Perimeter (ABCD) = 38
> > >   *  Let EF = x, EH = y\n  *  Perimeter (EFGH) = 2 * (x + y) = 38
> > > 3.  **Given EF:**
> > >   *  EF = x = 10
> > > 4.  **Solve for EH:**
> > >   *  2 * (10 + y) = 38
> > >   *  10 + y = 19
> > >   *  y = 19 - 10 = 9
> > >  **Answer:**
> > >  The length of the EH side of the EFGH parallelogram is 9.
> > > ```
> > > Example 2:
> > > ```
> > > Question: William is designing a sports field shaped like the cyan parallelogram in the following image <image1>. They plan to mark the boundary around the perimeter using synthetic turf. The price per unit length is given in the following figure <image2>. What would be the total cost for this material? All the computations are rounded to two places of decimal.\n You should provide a detailed reasoning process and justification for your answer.\n
> > >
> > > Ground truth answer: 128.00
> > >
> > > BlindSight answer:
> > > ### **Step 1: Identify the shape and dimensions**
> > >
> > > The sports field is shaped like a parallelogram. From the image:
> > >
> > > -   One pair of opposite sides has a length of 999 units.
> > > -   The other pair of opposite sides has a length of 777 units.
> > > -
> > > ### **Step 2: Calculate the perimeter of the parallelogram**
> > >
> > > The perimeter PPP of a parallelogram is given by the formula:
> > > P=2×(length of one pair of opposite sides+length of the other pair of opposite sides)
> > >
> > > Substituting the given values:
> > > P=2×(9+7)=2×16=32
> > > So, the perimeter of the parallelogram is 32 units.
> > >
> > > ### **Step 3: Determine the price of synthetic turf per unit length**
> > >
> > > From the table provided in the second image, the price of synthetic turf is 4currency units per unit length.
> > >
> > > ### **Step 4: Calculate the total cost**
> > >
> > > The total cost CCC is the product of the perimeter and the price per unit length:
> > > C=Perimeter×Price per unit length
> > > Substituting the values:
> > > C=32×4=128
> > >
> > > ### **Final Answer:**
> > >
> > > The total cost for marking the boundary with synthetic turf is:
> > > 128
> > > ```
> > >
> > > Additionally, the MMIE benchmark includes both multiple-choice and open-ended (generative) questions. We have provided accuracy results for a subset of the MMIE dataset that focuses on generative tasks. The examples above, along with these results, demonstrate that BlindSight preserves reasoning ability in open-ended tasks.
> > >
> > > ||Qwen2 VL (7B) | Qwen2.5 VL (7B)|
> > > |--|--|--|
> > > |Baseline|71.46|79.01|
> > > |BlindSight|71.18|78.83|

---

### Official Review · Reviewer_JbAX · 2025-10-31

**Soundness:** 3
**Presentation:** 3
**Contribution:** 2
**Rating:** 4
**Confidence:** 2

**Summary:**

This paper addresses the problem of long Time To First Token (TTFT) in large vision-language models (VLMs), which is caused by long prompt lengths resulting from the inclusion of images. The authors observe that the attention computation in VLMs processing multi-image inputs is inherently sparse, specifically noting the absence of cross-image attention in many layers. Based on this, the paper proposes "BlindSight," a training-free optimization approach. This method utilizes an input-template-aware attention sparsity mask to optimize VLM inference. Specifically, the authors use a dataset to derive a prompt-agnostic categorization for attention heads (Dense, Sink, Intra-Image, Intra-Image+Sink). Furthermore, a Triton-based GPU kernel is developed to leverage this sparsity. Experimental results show that BlindSight achieves a 1.8-3.2x speedup in attention computation for long prompts (36K-300K) with a reported average accuracy degradation of only 1.15% on the evaluated multi-image understanding benchmarks.

**Strengths:**

1. The paper's categorization of specific sparsity patterns (Intra-Image, Sink) in multi-image VLMs and linking this sparsity to modality boundary tokens (e.g., <image_start>) is a valuable insight.

2. BlindSight is a training-free method, which means it can be readily applied to existing pre-trained models without costly retraining, making it highly practical.

3. The paper goes beyond theoretical analysis by developing a custom Triton GPU kernel, demonstrating a clear path to translating this sparsity into real-world performance gains.

**Weaknesses:**

1. The paper's claim of an "average accuracy degradation of only 1.15%" is misleading. A closer look at Table 1 reveals significant performance drops on certain benchmarks. For example: On Qwen2.5-VL (32B), the MMIU benchmark drops from 44.67 to 41.49 (an absolute drop of 3.18 points, or ~7.1% relative degradation). On Gemma 3 (12B), the MUIRBench benchmark drops from 50.64 to 46.62 (an absolute drop of 4.02 points, or ~7.9% relative degradation). These are substantial performance hits that cannot be considered "minimal," suggesting the method is not robust in preserving performance.

2. The method relies heavily on per-image delimiter tokens in multi-image inputs. The authors admit this makes it inapplicable to current video processing schemes (which typically use only one pair of delimiters for the entire video). This is a major limitation, as video is a primary use case for long-context VLMs.

3. The method introduces several hyperparameters ($\alpha_{layer}$, $\gamma_{d}$, $\gamma_{s}$, $\gamma_{i}$) that require careful, model-specific tuning. For instance, Qwen and Gemma models require different (fixed vs. linear) $\alpha_{layer}$ strategies. This diminishes the "training-free" claim, as it necessitates a sensitive, model-specific tuning process on a representative dataset.

4. While the categorization of VLM sparsity patterns is useful, it builds heavily on existing work on attention sinks and static/dynamic sparsity patterns. Given the significant accuracy trade-off, this incremental novelty may not be sufficient.

**Questions:**

See above

---

> ### Author Response · Authors · 2025-11-23
> **Author Response**
>
> Thank you for taking the time to review our paper in detail! We appreciate your insightful questions and constructive feedback. Please take a look at our updated manuscript as well.
>
> > **The paper's claim of an "average accuracy degradation of only 1.15%" is misleading. A closer look at Table 1 reveals significant performance drops on certain benchmarks. For example: On Qwen2.5-VL (32B), the MMIU benchmark drops from 44.67 to 41.49 (an absolute drop of 3.18 points, or ~7.1% relative degradation). On Gemma 3 (12B), the MuirBench benchmark drops from 50.64 to 46.62 (an absolute drop of 4.02 points, or ~7.9% relative degradation). These are substantial performance hits that cannot be considered "minimal," suggesting the method is not robust in preserving performance.**
>
> Thank you for pointing this out. After reviewing the cases with higher degradation, we identified that BlindSight kernel used a hard-coded mask value `(-1e7)` instead of `-float(inf)`, causing underflow in specific scenarios. We corrected this and updated Table 1.
>
> With this optimization:
>
> -   **Qwen models show improved accuracy with BlindSight**, and
> -   **Gemma models improve slightly**, though they remain more sensitive due to sliding-window attention in certain layers and non-causal intra-image attention.
>
> BlindSight now achieves close-to-baseline accuracy across most benchmarks, with worst-case drops reduced significantly. For tasks where baseline models themselves underperform (e.g., MuirBench on Gemma 3 12B), sparsification has a larger impact.
>
> **Changes in the revised version:** Updated Table 1 with new accuracy results. We have clarified that the accuracy drop is absolute.
>
> > **The method relies heavily on per-image delimiter tokens in multi-image inputs. The authors admit this makes it inapplicable to current video processing schemes (which typically use only one pair of delimiters for the entire video). This is a major limitation, as video is a primary use case for long-context VLMs.**
>
> BlindSight currently depends on per-image (or per-frame) delimiters, which limits applicability to video inputs without such markers.  An important motivation for our paper was to encourage future VLM designers to introduce delimiter tokens to video frames and leverage sparsity to improve training time. We are excited to note that Qwen3-VL has already made strides in this direction by introducing per-frame delimiters and text-based timestamps during video processing. As a result, we observe sparsity in video inputs similar to multi-image inputs, as highlighted in Appendix K. Extending BlindSight to video inputs is a key direction for future work.
>
> **Changes in the revised version:** Added Appendix K to include attention sparsity patterns observed for video inputs in Qwen3-VL.
>
> > **The method introduces several hyperparameters (, , , ) that require careful, model-specific tuning. For instance, Qwen and Gemma models require different (fixed vs. linear) strategies. This diminishes the "training-free" claim, as it necessitates a sensitive, model-specific tuning process on a representative dataset.**
>
> We recognize the reviewer's concern regarding multiple parameters, but tuning is lightweight and does not involve retraining or inference-time computation. We recommend users use $\alpha_{layer}=0.1$, $\gamma_d=0.25$ and $\gamma_s=\gamma_i=0.6$ for typical VLM architectures similar to the Qwen series and Llama 4. We provide a detailed ablation study in Appendix D and show that mainly $\alpha_{layer}$ and $\gamma_{d}$ impact the accuracy with a clear trend for accuracy-sparsity trade-off.
>
> The Gemma model represents an unusual case, given its non-causal intra-image attention and sliding window attention in certain layers. The linear alpha scheme was derived to obtain optimal performance, though, as shown in Appendix D.1.2, performance degradation is not drastic even without specialization. We also showcase the robustness of these parameter combinations by characterizing attention heads on an alternate dataset (Appendix F). Per your feedback, we have updated the abstract to clarify that BlindSight does not require additional inference-time overhead.
>
> **Changes in the revised version:** Expanded Appendix D with more hyperparameter ablation studies, added Appendix F to showcase the applicability of the suggested hyperparameters for different characterization datasets.

---

> ### Author Response · Authors · 2025-11-23
> **Author Response**
>
> > **While the categorization of VLM sparsity patterns is useful, it builds heavily on existing work on attention sinks and static/dynamic sparsity patterns. Given the significant accuracy trade-off, this incremental novelty may not be sufficient.**
>
> We believe BlindSight is the first technique to uncover prompt-structure dependent sparsity patterns unique to multi-image scenarios. While we build on the core observation that attention sinks develop in VLMs, we highlight empirical evidence in Section 7 showcasing their unique role in VLMs. Several prior works have focused on attention sinks within a single image or relied on online metrics such as attention score to key tokens in VLMs. Ours is the first work to demonstrate that inter-image attention occurs through fixed attention sinks that can be characterized offline. Given that video models are trending toward similar sparsity patterns (Appendix K), BlindSight's insights and kernels can serve as key components for future efficient VLM architectures.
>
> Thank you again for your thorough analysis. We hope that our updated manuscript addresses your comments and increases confidence in our work.

---

> ### Comment · Reviewer_JbAX · 2025-11-25
> **Reply to authors**
>
> Thanks for your reply. As you reply in Question 1, there is some issues in the implementation (hard-coded mask value) of your original method. And your updated results can not convince me. Therefore, this submission is not ready to current conference, I will keep my score.

---

> > ### Author Response · Authors · 2025-12-03
> > **Author response**
> >
> > Thank you for your follow-up and for carefully reviewing our work. We understand and respect your decision to maintain your score. Below, we summarize our responses to your comments and concerns:
> >
> > -   **Optimization of BlindSight kernel for accuracy improvement** Improved performance of the BlindSight kernel by updating the occurrence of large negative numbers with the Triton in-built $-\inf$ value. With this optimization, the updated BlindSight kernel now achieves near-baseline accuracy across most benchmarks (updated Table 1).
> > -   **Video Support:** Added sparsity patterns for video inputs using Qwen3-VL, highlighting BlindSight’s applicability to future video-based VLMs.
> > -   **Hyperparameter Analysis:** Expanded ablation studies, identified key parameters, and provided default recommendations for their values. We have evaluated these default values on a new model (Qwen3-VL) and an alternative characterization dataset. Notably, only one hyperparameter ($\alpha_{layer}$) is used during characterization, while the others are applied during a lightweight aggregation step, significantly reducing the overhead of tuning.
> >
> > We believe these revisions substantially strengthen the paper and address the core concerns raised. Thank you again for your constructive feedback—it has helped us improve the clarity and rigor of our work.

---

### Official Review · Reviewer_3Um8 · 2025-11-03

**Soundness:** 2
**Presentation:** 2
**Contribution:** 2
**Rating:** 6
**Confidence:** 3

**Summary:**

BlindSight is a training-free method that accelerates multi-image vision-language model inference by leveraging the inherent sparsity in attention computation, constructing input-template-aware sparse masks without modifying model architecture. It delivers 1.8–3.2× attention speedup and only about 1.15% accuracy degradation on major benchmarks for Qwen2-VL, Qwen2.5-VL, and Gemma 3 models

**Strengths:**

BlindSight offers significant inference acceleration for multi-image vision-language models by exploiting attention sparsity without requiring extra training or changes to model architecture.

It maintains almost the same accuracy as dense attention, showing an average accuracy degradation of only about 1.15% across major benchmarks.

**Weaknesses:**

BlindSight relies on predefined sparsity patterns, so it may not capture context-dependent attention dynamics that could be important for some prompts or tasks.

The minimal accuracy drop is measured only on major benchmarks; specific cases or other domains might experience higher accuracy degradation.

Integration requires careful attention boundary detection, and underlying model changes (e.g., image tokenization strategy) may affect its effectiveness or compatibility.

**Questions:**

Here are some possible questions for the paper “BlindSight: Harnessing Sparsity for Efficient Vision-Language Models”:

How well does the BlindSight approach generalize to vision-language models beyond those tested, such as proprietary or non-transformer architectures?

Could the predefined sparse mask templates miss specialized cross-modal or long-range interactions in exceptional tasks, and how can adaptability be improved?

What are the trade-offs when tuning the sparsity and accuracy thresholds, and how robust are these methods across varying image resolutions and prompt structures?

---

> ### Author Response · Authors · 2025-11-23
> **Author response**
>
> Thank you for your review and insightful questions! We hope that our updated paper addresses your concerns and questions.
>
> > **BlindSight relies on predefined sparsity patterns, so it may not capture context-dependent attention dynamics that could be important for some prompts or tasks.**
>
> BlindSight is a task-agnostic technique that exploits the inherent sparsity in VLMs that develops during the training phase. From our observations, we believe that once attention heads specialize into dense and sparse types for most prompts, task-specific long-range interactions are conducted through dense layers and attention sinks. BlindSight's performance is shown to be robust across diverse evaluation benchmarks (Section E lists different tasks evaluated) shown in Table 1. Prior works on leveraging  Static Attention Sparsity in LLM attention layer follow a similar approach (Section 2) where offline characterization was used to reveal inherent sparsity that can be leveraged in a task-agnostic manner.
>
> > **The minimal accuracy drop is measured only on major benchmarks; specific cases or other domains might experience higher accuracy degradation.**
>
> This is a common concern across many post-training optimization techniques. We agree that domain-specific prompts may exhibit different sparsity characteristics. BlindSight mitigates this risk through two mechanisms:
>
> - **Prompt-level NMSE thresholding** (Algorithm 1 Section 4.1), which reverts to dense attention when sparsification could harm accuracy; and
>
> - **Dataset-level aggregation** (Algorithm 2 Section 4.2), ensuring that only dominant patterns are applied globally. Our recommendation for BlindSight users is to include domain-specific samples to ensure coverage in this step
>
> While BlindSight is a post-training optimization, our goal is to also uncover inherent redundancy in attention computation in VLMs. We encourage model builders to design hybrid dense-sparse VLM architectures inspired by BlindSight and train models from scratch using attention kernels with limited inter-image attention.
>
> > **Integration requires careful attention boundary detection, and underlying model changes (e.g., image tokenization strategy) may affect its effectiveness or compatibility.**
>
> The presence of distinct boundary tokens  has become a feature of most popular VLMs today (Qwen series, Llama 4, Gemma series). We expect this trend of models adopting a per-image delimiter to continue. The special tokens used as delimiter are typically defined by prompt processor configuration.
>
> In the paper, we specifically focus on two categories of models, a typical Qwen-like model and an atypical Gemma model. While the tokenization schemes vary, we continue to observe BlindSight-like sparsity even in newer models such as **Qwen3** or **Llama 4** (Figure 6, Appendix A).
>
> We show in Appendix G how BlindSight consistently provides close to baseline accuracy for varying resolution of the images (i.e., different number of tokens/image). Finally, we also observe that this sparsity persists in the presence of token pruning (Appendix J), strengthening our insight that significant sparsity depends on prompt structure rather than token values in VLMs that can be leveraged through offline characterization without runtime overhead.
>
> **Changes in the revised version:** Added Appendix A to include attention sparsity models for additional models. Added Appendix G and Appendix J to show BlindSight’s accuracy across different image resolutions and under token compression.
>
> > **How well does the BlindSight approach generalize to vision-language models beyond those tested, such as proprietary or non-transformer architectures?**
>
> BlindSight is designed for transformer-based VLMs, which dominate current architectures. Most open-source and proprietary VLMs such as Qwen, Llama, and Gemma families, use per-image delimiters and vision transformer backbones, enabling BlindSight’s applicability. We also observed similar sparsity patterns in newer models such as Qwen3 and Llama 4 (Appendix A), suggesting that these patterns are architecture-agnostic within transformer designs.
>
> We also note that BlindSight is specifically applicable to transformer layers in VLMs. For non-transformer architectures, BlindSight’s current implementation would not apply directly; however, its core insight on leveraging modality boundary-induced sparsity can inspire analogous optimizations.
>
> **Changes in the revised version:** Added attention sparsity patterns for Qwen 3 and Llama 4 in Appendix A,

---

> ### Author Response · Authors · 2025-11-23
> **Author Response**
>
> > **Could the predefined sparse mask templates miss specialized cross-modal or long-range interactions in exceptional tasks, and how can adaptability be improved?**
>
> BlindSight proposes different types of attention masks to leverage inter-image and intra-image attention sparsity, and uses offline characterization to find which attention pattern is suitable for a given head. Due to this, different attention heads are assigned different types of attention masks, including dense attention for some heads. We believe that different heads retrieve different information (long range, local, cross-modal) and by characterizing and assigning the sparsity mask type, we sparsify heads that do not perform such long range interaction.  In case of scenarios where performance losses are still observed, we would expect that performance can be recovered by fine-tuning the model with sparse attention masks.
>
> > **What are the trade-offs when tuning the sparsity and accuracy thresholds, and how robust are these methods across varying image resolutions and prompt structures?**
>
> We study the impact of hyperparameters on the sparsity/accuracy in Appendix D. Lower $\alpha_{layer}$ and lower $\gamma_d​$ increase the number of dense heads, preserving accuracy but reducing FLOPs savings; conversely, higher sparsity thresholds yield greater speedups at the cost of accuracy. For example, increase $\alpha_{layer}$ from $0.05$ to $0.2$ increases the number of sparse heads from $40\%$ to $92\%$ at the cost of $5\%$ absolute reduction in accuracy (Appendix D.1.1). We further show FLOPs reduction versus accuracy for different values of $\alpha_{layer}$ in Appendix I (Figure 12). As $\alpha_{layer}$ decreases, FLOPs reduction decreases and accuracy improves. The accuracy saturates (knee point) at a point, which provides the best trade-off between FLOPs reduction and accuracy.
>
> We have also added a study showcasing the robustness of BlindSight to image resolution in Appendix G. We do not observe any impact of image resolution on the effectiveness of BlindSight and it consistently achieves accuracy close to dense baseline.
>
> We have added details on different multi-image tasks evaluated in this work in Appendix E. Different tasks result in different prompt structures (e.g., image resolution, number of images, text and image placement and interleaving).  BlindSight performs comparably in many multi-image analysis benchmarks evaluated in Table 1, each of them comprising of various prompt structure.
>
> **Changes in the revised version:** Expanded Appendix D (impact of hyperparameters on accuracy and sparsity), introduced Appendix I (FLOPs-accuracy trade-off).  Introduced Appendix E (dataset details) and Appendix G (Impact of image resolution).
>
>
> Thank you again for your valuable feedback. We hope our clarifications adequately address your questions and strengthen your confidence in our work.

---

### Author Response · Authors · 2025-12-03
**Summary of Revisions and Strengthened Contributions**

We thank the reviewers for their thorough review, highlighting opportunities to strengthen BlindSight with additional experimental evidence. With this paper, we propose a novel post-training VLM optimization scheme that achieves significant performance improvements while maintaining accuracy.

Below we summarize major changes and additional results we have included based on the reviewers' feedback and questions. These additions directly address reviewer concerns regarding robustness, generalization, and practical impact.

**Expanded Evaluation for Robustness and Generalization**

-   **Appendix A:** Sparsity patterns for new models (Qwen3, Llama 4).
-   **Appendix K:** Sparsity patterns for video inputs (Qwen3-VL).
-   **Appendix G**: Robustness to image resolution.
-   **Appendix J**: Behavior under token compression (DivPrune).
-   **Appendix F**: Robustness to choice of characterization dataset.
-   Added performance results for Qwen3-VL (32B) demonstrating BlindSight’s applicability with default hyperparameters (rebuttal response comment).

**Hyperparameter Ablation Studies**

-   **Appendix D.1.2**: Impact of linear vs. fixed alpha for Gemma 3.
-   **Appendix I**: Accuracy–FLOPs trade-off analysis showing clear trends and knee points for alpha.
-   **Section 6.1**: Added recommendation on default values of hyperparameters and highlighted key hyperparameters and how their values impact accuracy and compute requirement.
-   Additional results for Qwen3-VL (32B) confirming robustness of default hyperparameters (rebuttal response comment).

**Accuracy-Compute trade-off:**

- **Appendix H**: FLOPs reduction results to complement the accuracy results in Table 1.
-  **Appendix I**: Accuracy-FLOPs trade-off results for varying values of alpha. This Section also highlights that there is a clear trend between the values of alpha and accuracy, with a knee-point beyond which accuracy saturates.

**Comparison with token compression:**

- **Appendix J**: We have provided qualitative and quantitative comparison with token compression.
- **Appendix J**: We have also included sparsity patterns for token compression and accuracy results for DivPrune + BlindSight to highlight how BlindSight can be combined with token compression to further reduce computation.

**Accuracy numbers revision**:

- Modified Table 1 with updated BlindSight kernel and added comparison with fixed mask type baseline to highlight the value of BlindSight's sparsity selection approach for different attention heads.

We have uploaded our manuscript with a complete set of experimental evaluations. We thank you again for your time and feedback! We believe these comprehensive updates not only address all reviewer concerns but also position BlindSight as a practical and scalable solution for future VLM architectures.

---

### Meta-Review · Area_Chair_3FDH · 2026-01-08

**Summary:**

BlindSight is a training-free inference optimization for multi-image VLMs by studying the attention pattern and they reported ~1.8–3.2× attention speedup with small average accuracy loss. Reviewers generally like the practicality (no retraining) and the kernel-level realization of speedups, plus the characterization of modality-boundary/sink effects. There are couples of related works missing in the draft raised by the reviewers.

[FastV] https://arxiv.org/abs/2403.06764

[LLaVA-PruMerge] https://arxiv.org/abs/2403.15388

[DivPrune] https://arxiv.org/abs/2503.02175

[DART] https://arxiv.org/abs/2502.11494

[FastVLM] Efficient Vision Encoding for Vision Language Models.”, CVPR 2025

Reviewers shared concerns on the experiments (i.e. beyond multiple choice short answer, more baseline comparison, video). We encourage the authors to enrich the draft with those experiments for next submission.

**Reviewer Concerns:**

(Partially) addressed: accuracy drop concern, more comparisons with related works, video

Outstanding: FastVLM comparison and generalization issue

**Reviewer Scores:**

Reviewers has updated scores after discussion.

---

### Decision · Program_Chairs · 2026-01-26

Reject